# Transient regulation of focal adhesion via Tensin3 is required for nascent oligodendrocyte differentiation

Emeric Merour[1†], Hatem Hmidan[1†‡], Corentine Marie[1], Pierre-Henri Helou[1], Haiyang Lu[1], Antoine Potel[1], Jean-Baptiste Hure[1], Adrien Clavairoly[1], Yi Ping Shih[2], Salman Goudarzi[3], Sebastien Dussaud[1], Philippe Ravassard[1], Sassan Hafizi[3], Su Hao Lo[2], Bassem A Hassan[1], Carlos Parras[1]*

[1]Paris Brain Institute, Sorbonne Université, Inserm U1127, CNRS UMR 7225, Hôpital Pitié-Salpêtrière, Paris, France; [2]Department of Biochemistry and Molecular Medicine, University of California, Davis, Davis, United States; [3]School of Pharmacy and Biomedical Sciences, University of Portsmouth, Portsmouth, United Kingdom

*For correspondence:
carlos.parras@icm-institute.org

[†]These authors contributed equally to this work

Present address: [‡]Physiology and Pharmacology Department, College of Medicine, Al-Quds University, Jerusalem, Palestine

Competing interest: The authors declare that no competing interests exist.

**Abstract** The differentiation of oligodendroglia from oligodendrocyte precursor cells (OPCs) to complex and extensive myelinating oligodendrocytes (OLs) is a multistep process that involves large-scale morphological changes with significant strain on the cytoskeleton. While key chromatin and transcriptional regulators of differentiation have been identified, their target genes responsible for the morphological changes occurring during OL myelination are still largely unknown. Here, we show that the regulator of focal adhesion, Tensin3 (Tns3), is a direct target gene of Olig2, Chd7, and Chd8, transcriptional regulators of OL differentiation. Tns3 is transiently upregulated and localized to cell processes of immature OLs, together with integrin-β1, a key mediator of survival at this transient stage. Constitutive *Tns3* loss of function leads to reduced viability in mouse and humans, with surviving knockout mice still expressing Tns3 in oligodendroglia. Acute deletion of *Tns3* in vivo, either in postnatal neural stem cells (NSCs) or in OPCs, leads to a twofold reduction in OL numbers. We find that the transient upregulation of Tns3 is required to protect differentiating OPCs and immature OLs from cell death by preventing the upregulation of p53, a key regulator of apoptosis. Altogether, our findings reveal a specific time window during which transcriptional upregulation of Tns3 in immature OLs is required for OL differentiation likely by mediating integrin-β1 survival signaling to the actin cytoskeleton as OL undergo the large morphological changes required for their terminal differentiation.

## Editor's evaluation

This work provides a major contribution to the field of oligodendrogenesis. It is a comprehensive analysis of Tensin3 function and is conceptually novel as it links the fields of transcriptional control in oligodendrocyte lineage cells with morphogenetic changes and integrin signaling-mediated cell survival. Finally, it is also the discovery of a very useful marker for remyelinating oligodendrocytes in disease conditions.

## Introduction

Oligodendrocyte (OL) lineage cells, mainly constituted by oligodendrocyte precursor cells (OPCs) and OLs, play key roles during brain development and neuronal support by allowing saltatory conduction of myelinated axons and metabolically supporting these axons with lactate or glucose shuttling

through the myelin sheath (*Fünfschilling et al., 2012*; *Lee et al., 2012*; *Meyer et al., 2018*). Accumulating evidence also indicates their fundamental contribution to different aspects of adaptive myelination, a type of brain plasticity (*Mount and Monje, 2017*; *Yang et al., 2020*), shown by the requirement of new oligodendrogenesis for proper learning and memory in motor, spatial, and fear-conditioning learning paradigms (*McKenzie et al., 2014*; *Xiao et al., 2016*; *Steadman et al., 2019*; *Pan et al., 2020*; *Wang et al., 2020*; *Xin and Chan, 2020*). Furthermore, oligodendroglia and myelin pathologies have been recently linked not only to the development of glioma (*Liu et al., 2011*) but to developmental (*Castelijns et al., 2020*; *Phan et al., 2020*), neurodegenerative (*Grubman et al., 2019*; *Bryois et al., 2020*), and psychiatric (*Nott et al., 2019*) diseases.

Unlike most precursor cells, OPCs constitute a stable population of the postnatal and adult central nervous system (CNS) (*Ffrench-Constant and Raff, 1986*; *Suzuki and Goldman, 2003*). Therefore, OPCs need to keep a tight balance between proliferation, survival, and differentiation. This balance is crucial to maintain the OPC pool while contributing to myelin plasticity in adult life and to remyelination in diseases such as multiple sclerosis (MS). Demyelinated MS plaques can be normally repaired in early stages of the disease, presumably by endogenous OPCs, but this repair process becomes increasingly inefficient with aging, when OPC differentiation seems to be partially impaired (*Chang et al., 2002*; *Compston and Coles, 2002*; *Neumann et al., 2019*). Therefore, understanding the mechanisms involved in OPC differentiation is critical to foster successful (re)myelination in myelin pathologies.

A large diversity of extrinsic signals, including those mediated by integrin signaling (reviewed in *Bergles and Richardson, 2016*), as well as many intrinsic factors, including transcription factors (TFs) and chromatin remodelers (reviewed in *Emery and Lu, 2015*; *Parras et al., 2020*), are involved in OPC proliferation, survival, and differentiation. However, the mechanisms for how these signals balance OPC behavior is largely unknown. OPC differentiation requires profound changes in chromatin and gene expression (*Emery and Lu, 2015*; *Küspert and Wegner, 2016*; *Wheeler and Fuss, 2016*). TFs, such as Olig2, Sox10, Nkx2.2, or Ascl1, are key regulators of OL differentiation by directly controlling the transcription of genes implicated in this process (*Qi et al., 2001*; *Stolt et al., 2002*; *Nakatani et al., 2013*; *Yu et al., 2013*), but being already expressed at the OPC stage, it is still unclear how these TFs control the induction of differentiation. A growing body of evidence suggests that some of these TFs work together with chromatin remodeling factors during transcriptional initiation/elongation to drive robust transcription (*Zaret and Mango, 2016*). Accordingly, Olig2 and Sox10 TFs have been shown to cooperate with chromatin remodelers such as Brg1 (*Yu et al., 2013*), Chd7 (*He et al., 2016*; *Marie et al., 2018*), Chd8 (*Marie et al., 2018*; *Zhao et al., 2018*), and EP400 (*Elsesser et al., 2019*), to promote the expression of OL differentiation genes. To improve our understanding of the mechanisms of OL differentiation, we searched for novel common targets of these key regulators by generating and analyzing the common binding profiles of Olig2, Chd7, and Chd8 in gene regulatory elements of differentiating oligodendroglia. We identified *Tns3*, coding for the focal adhesion protein Tensin3, as one such target and showed that it is expressed in immature OLs (iOLs) during myelination and remyelination, thus constituting a marker for this transient oligodendroglial stage. Using different genetic strategies to induce *Tns3* loss-of-function mutations in vivo, we describe the function of a Tensin family member in the CNS, demonstrating that Tns3 is required for OL differentiation in the postnatal mouse brain, at least in part by mediating integrin-β1 signaling, essential for survival of differentiating oligodendroglia (*Colognato et al., 2002*; *Benninger et al., 2006*).

## Results

### *Tns3* is a direct target gene of key regulators of oligodendrocyte differentiation

To find new factors involved in OL differentiation, we screened for target genes of Olig2, Chd7, and Chd8, key regulators of oligodendrogenesis (*Lu et al., 2000*; *Lu et al., 2002*; *Yu et al., 2013*; *He et al., 2016*; *Marie et al., 2018*; *Zhao et al., 2018*; *Parras et al., 2020*). We generated and compared the genome-wide binding profiles for these factors in acutely purified oligodendroglial cells from postnatal mouse brain cortices by magnetic cell sorting (MACS) of O4$^+$ cells (*Marie et al., 2018*). MACS-purified cells, composed of 80% PDGFRα$^+$ OPCs and 20% of Nkx2.2$^+$/CC1$^+$ iOLs, were subjected to chromatin immunoprecipitation followed by sequencing (ChIP-seq) for Olig2 and histone

modifications marking the transcription activity of gene regulatory elements (H3K4me3, H3K4me1, H3K27me3, and H3K27ac; *Figure 1a*). The profile of activity histone marks at Olig2-binding sites indicated that Olig2 binds promoters (60%) and enhancers (40%) with either active or more poised/repressive states (*Figure 1—figure supplement 1a–f*), supporting the suggested pioneer function of Olig2 in oligodendrogenesis (*Yu et al., 2013*). Among the 16,578 chromatin sites bound by Olig2 corresponding to 8672 genes (*Figure 1—figure supplement 1d*), there were key regulators of OL differentiation, including *Ascl1, Sox10, Myrf, Chd8,* and *Smarca4/Brg1* (*Figure 1b*; *Supplementary file 1*). Combining Olig2 with Chd7 and Chd8 binding profiles, which we previously generated using the same protocol (*Marie et al., 2018*), we found 1774 chromatin sites commonly bound by the three regulators, with half of them (47% and 832 peaks) corresponding to active promoters (H3K4me3/H3K27ac marks) of 654 protein-coding genes (*Figure 1c*, *Supplementary file 1*). Among these genes, *Tns3* coding for Tensin3, a focal adhesion protein deregulated in certain cancers (*Martuszewska et al., 2009*), showed the highest expression levels in iOLs relative to other brain cell types (*Zhang et al., 2014*; *Figure 1d*). Indeed, Olig2, Chd7, and Chd8 commonly bound three putative promoters of *Tns3* having active transcription marks in purified oligodendroglia (H3K27ac/H3K24me3; *Figure 1e*). To directly assess whether *Tns3* expression requires the activity of these key regulators, we interrogated the transcriptomes of these oligodendroglial cells purified from *Chd7iKO* (*Pdgfra-CreER^T; Chd7^flox/flox*), *Chd8cKO* (*Olig1^Cre; Chd8 ^flox/flox*), and their respective control cortices (*Marie et al., 2018*; *Zhao et al., 2018*). *Tns3* transcripts were largely downregulated upon acute deletion of these factors in postnatal OPCs/iOLs (*Figure 1f and g*), indicating that *Tns3* expression in OPCs/iOLs is directly controlled by Chd7 and Chd8 chromatin remodelers, key regulators of OL differentiation.

## *Tns3* transcripts are highly expressed in mouse and human immature oligodendrocytes

We then investigated *Tns3* expression pattern in the brain. High expression levels of *Tns3* transcripts in iOLs, compared to its low expression in other cells of the postnatal mouse brain detected by bulk transcriptomics (*Figure 1—figure supplement 2a*; *Zhang et al., 2014*), were paralleled by the sparse labeling with *Tns3* probes enriched in the white matter of the postnatal and adult brain detected by in situ hybridization (*Figure 1—figure supplement 2b*; Allen Brain Atlas, https://portal.brain-map.org/). By harnessing single-cell transcriptomics (scRNA-seq), we sought to create an integrative gene profiling for OL lineage cells by bioinformatics integration and analyses of OL lineage cells at embryonic, postnatal, and adult stages (*Marques et al., 2016*). We thus integrated these datasets using Seurat (*Stuart et al., 2019*) and selected 5516 progenitor and oligodendroglial cells. Unsupervised clustering and visualization of cells in two dimensions with uniform manifold approximation and projection (UMAP) identified nine different clusters following a differentiation trajectory. Based on known cell-subtype-specific markers (*Figure 1—figure supplement 2e* and *Supplementary file 1*), we could identify these clusters as (*Figure 1—figure supplement 2c and e*): (1) two types of neural stem/progenitor cells, which we named NSCs and NPCs according to their expression of stem cell (*Vim, Hes1, Id1*) and neural progenitor (*Sox11, Sox4, Dcx*) markers; (2) OPCs expressing their known markers (*Pdgfra, Cspg4, Ascl1*) and cycling OPCs also enriched in cell cycle markers (*Mki67, Pcna, Top2*); (3) two stages of iOLs, both expressing the recently proposed markers *Itpr2* and *Enpp6* (*Marques et al., 2016*; *Xiao et al., 2016*), and which are split by the expression of *Nkx2-2* (iOL1 being *Nkx2-2^+* and iOL2 being *Nkx2-2^-*), in agreement with our previous characterization by immunofluorescence (*Nakatani et al., 2013*; *Marie et al., 2018*); (4) myelin-forming oligodendrocytes (MFOLs), enriched in markers such as *Slc9a3r2* and *Igsf8*; and (5) two clusters of myelinating OLs, which we named MOL1 and MOL2, expressing transcripts of myelin proteins (*Cnp, Mag, Mbp, Plp1, Mog*) and some specific markers of each cluster, including *Mgst3, Pmp22* for MOL1 (corresponding to MOL1/2/3/4 clusters of *Marques et al., 2016*), and *Neat1, Grm3, Il33* for MOL2 (corresponding to MOL5/6 clusters of *Marques et al., 2016*). Interestingly, *Tns3* transcripts were strongly expressed in both iOL1 and iOL2 clusters (*Figure 1—figure supplement 2d and e*), similar to the recently proposed iOL markers *Itpr2* and *Enpp6* (*Figure 1—figure supplement 2e*), and downregulated in mature/terminally differentiated OLs, indicating that high levels of *Tns3* expression are specific to iOLs. We finally assessed whether *Tns3* expression pattern was conserved in human oligodendroglia pursuing a similar bioinformatics analysis using single-cell transcriptomes from human oligodendroglia differentiated from embryonic stem cells (*Chamling et al., 2021*). Upon integration with Seurat and

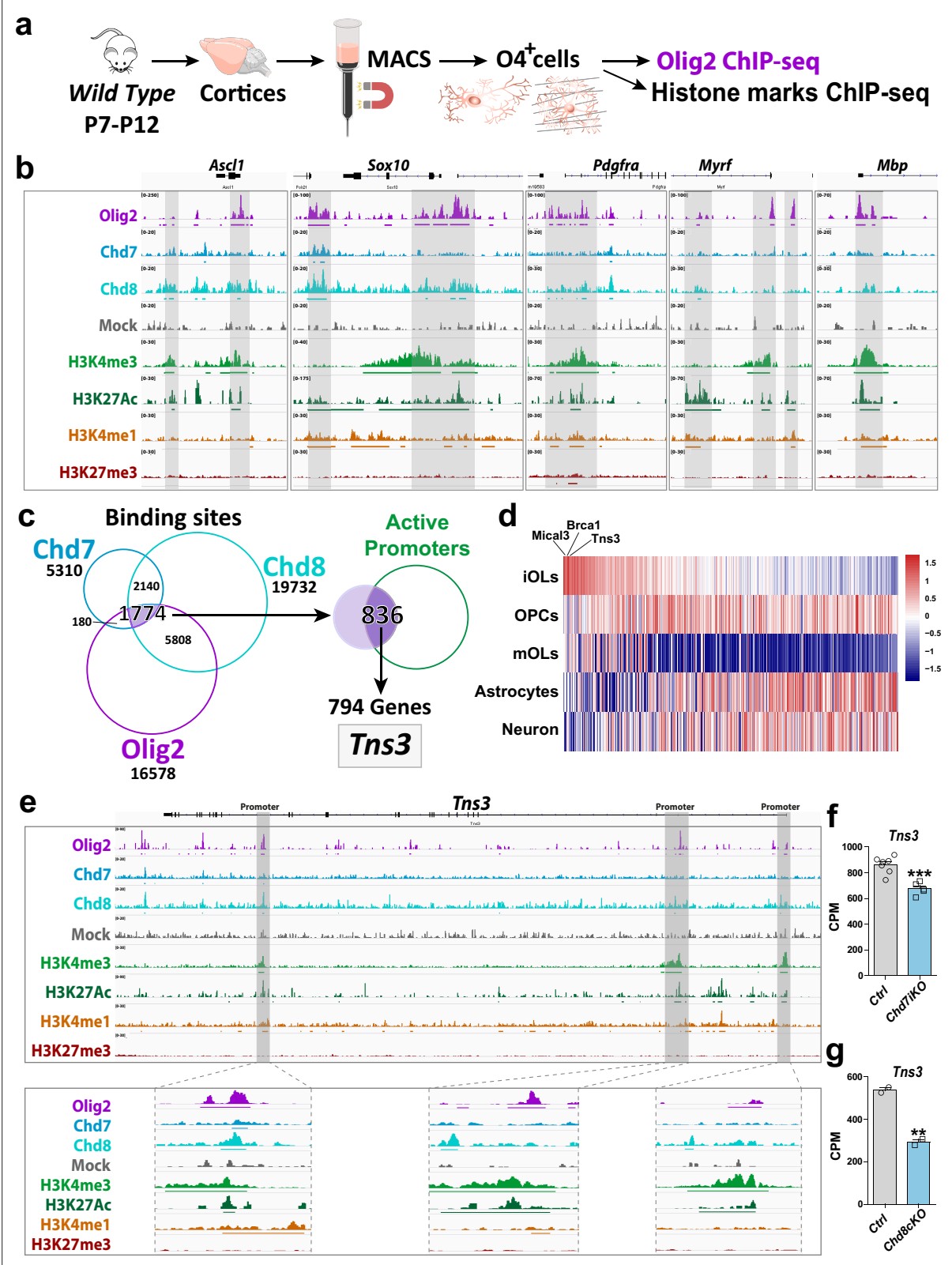

**Figure 1.** *Tns3* is a target gene of Olig2 and Chd7/8 regulators of oligodendrocyte differentiation. (**a**) Scheme representing MACSorting of O4+ cells from wild-type cortices followed by ChIP-seq for Olig2 and histone marks (H3K4me3, H3K27Ac, H3K4me1, and H3K27me3). (**b**) Tracks from IGV browser of *Ascl1*, *Sox10*, *Pdgfra*, *Myrf*, and *Mbp* gene regions depicting ChIP-seq data in O4+ cells (OPCs/OLs) for transcription factor Olig2, chromatin remodelers Chd7 and Chd8, and epigenetic marks (H3K4me3, H3K27ac, H3K4me1, and H3K27me3). Mock (control IgG) shows no peaks in the regions

*Figure 1 continued on next page*

*Figure 1 continued*

of interest. Lines present below peaks indicate statistical significance (by peak calling). (**c**) Strategy used to identify *Tns3* as a gene target of Olig2, Chd7, and Chd8, potentially involved in oligodendrogenesis. Left: Venn diagrams depicting the overlap of binding peaks between Chd7 (blue), Chd8 (cyan), and Olig2 (purple) in O4$^+$ cells. Right: Venn diagram showing that 836 (47%) of the 1774 common regions have marks of active promoters, corresponding to 794 genes, including *Tns3*. (**d**) Heatmap representing the expression of the 794 genes in immature oligodendrocytes (iOLs) compared to oligodendrocyte precursor cells (OPCs), myelinating oligodendrocytes (mOLs), astrocytes, and neurons. *Tns3* is the third most specific (data from **Zhang et al., 2014**). (**e**) Tracks from IGV browser of *Tns3* gene region depicturing ChIP-seq data in O4$^+$ cells (OPCs/OLs) for transcription factor Olig2 and epigenetic marks (H3K4me3, H3K27ac, H3K4me1, and H3K27me3), zooming in Tns3 alternative promoters. Mock (control IgG) shows no peaks in the regions of interest. Horizontal lines present below peaks indicate statistical significance (peak calling). (**f, g**) Barplots showing *Tns3* transcript count per million (CPM) in O4$^+$ cells upon tamoxifen-induced *Chd7* deletion (*Chd7iKO*), (**f**) or *Chd8* deletion (*Chd8cKO*), (**g**) compared to control (Ctrl). Statistics done using edgeR suite.

The online version of this article includes the following figure supplement(s) for figure 1:

**Figure supplement 1.** *Tns3* is a target gene of Olig2 and Chd7/8 regulators of oligodendrocyte differentiation.

**Figure supplement 2.** Strong expression of *Tns3* transcripts in mouse and human immature oligodendrocytes.

**Figure supplement 3.** Strong expression of *Tns3* transcripts in immature oligodendrocytes (iOLs) of human fetal cerebellum.

identification of cluster cell types using specific markers, we selected 7690 progenitor and oligodendroglial cells that corresponded to six main cluster cell types following a differentiation trajectory from neural cells (NSCs) up to iOLs (iOL1 and iOL2), as depicted by UMAP representation (*Figure 1—figure supplement 2f and h*). Cells expressing high levels of *TNS3* corresponded to iOLs (iOL1 and iOL2 clusters, *Figure 1—figure supplement 2g and h*). We obtained similar results analyzing a human fetal midterm cerebellum (GW9-GW22) dataset (*Aldinger et al., 2021*), with high levels of *TNS3* in iOLs similar to other suggested iOL markers such as *ITPR2*, *ENPP6*, and *BCAS1*, indicating a conserved expression pattern of Tns3/TNS3 between mouse and human oligodendrogenesis (*Figure 1—figure supplement 3*).

## Tns3 protein is enriched in the cytoplasm and processes of immature oligodendrocytes

Given the high expression level of *Tns3* transcripts in iOLs, we characterized the Tns3 protein expression pattern in the postnatal brain using commercial and homemade Tns3-recognizing antibodies. Optimization of immunofluorescence protocols demonstrated signal in CC1$^+$ OLs in the postnatal brain with four different antibodies (P24, *Figure 2—figure supplement 1*). To our surprise, while all antibodies showed signal localized in the cytoplasm and main processes of CC1+ OLs (*Figure 2—figure supplement 1a–d*), one Tns3-recognizing antibody (Millipore) also presented a strong nuclear signal (*Figure 2—figure supplement 1d*) never reported for Tns3 localization in other tissues (such as lung, liver, and intestine) (*Katz et al., 2007*; *Nishino et al., 2012*; *Cao et al., 2015*). To better characterize Tns3 protein expression pattern and its subcellular localization, we generated a knock-in mice tagging the Tns3 C-terminal side with a V5-tag (*Tns3$^{Tns3-V5}$* mice) by microinjecting mouse zygotes with a single-strand oligodeoxynucleotide (ssODN) containing V5 sequence together with Cas9 protein and a gRNA targeting the stop codon region of *Tns3* ('Materials and methods'; *Figure 2—figure supplement 2a–c*). We first verified by immunofluorescence that Tns3-V5 protein in *Tns3$^{Tns3-V5}$* mice presented the expression pattern reported for Tns3 in the lung and the kidney (*Figure 2—figure supplement 2d and e*). We then characterized Tns3 protein expression in oligodendroglia using V5 antibodies, finding that Tns3 protein can be detected at high levels in the cytoplasm and main processes of CC1$^+$ iOLs but not in their nuclei (*Figure 2a*). Using an antibody recognizing Itpr2, a suggested iOL marker (*Marques et al., 2016*), we saw that Tns3 largely overlapped with Itpr2 (*Figure 2b*). Using Nkx2.2 and Olig1$^{cytoplamic}$ expression distinguishing iOL1 and iOL2, respectively, we found high levels of Tns3 in iOL1s (Nkx2.2$^+$/Olig1$^-$ cells) and a fraction of iOL2s (Nkx2.2$^-$/Olig1$^{cytoplamic}$ cells; *Figure 2c, i and j*), suggesting that Tns3 protein expression peaks in early iOLs. Comparison with Opalin protein localized in the cell body, processes, and myelin segments of OLs showed that Tns3 levels decreased with increasing levels of Opalin, with Tns3-V5 levels undetectable in myelinating OLs (i.e., Opalin$^+$/CC1$^+$ cells presenting myelinated segments; *Figure 2d*, arrowheads). We then performed Western blot analysis with anti-V5 antibodies in purified O4$^+$ cells from P7, P14, and P21 *Tns3$^{Tns3-V5}$* mouse brains to assess their specificity to recognize Tns3-V5, knowing that two Tns3 isoforms can be detected at the transcript level in the human brain (*Figure 2—figure supplement 2f*, GTEX project, gtexportal.

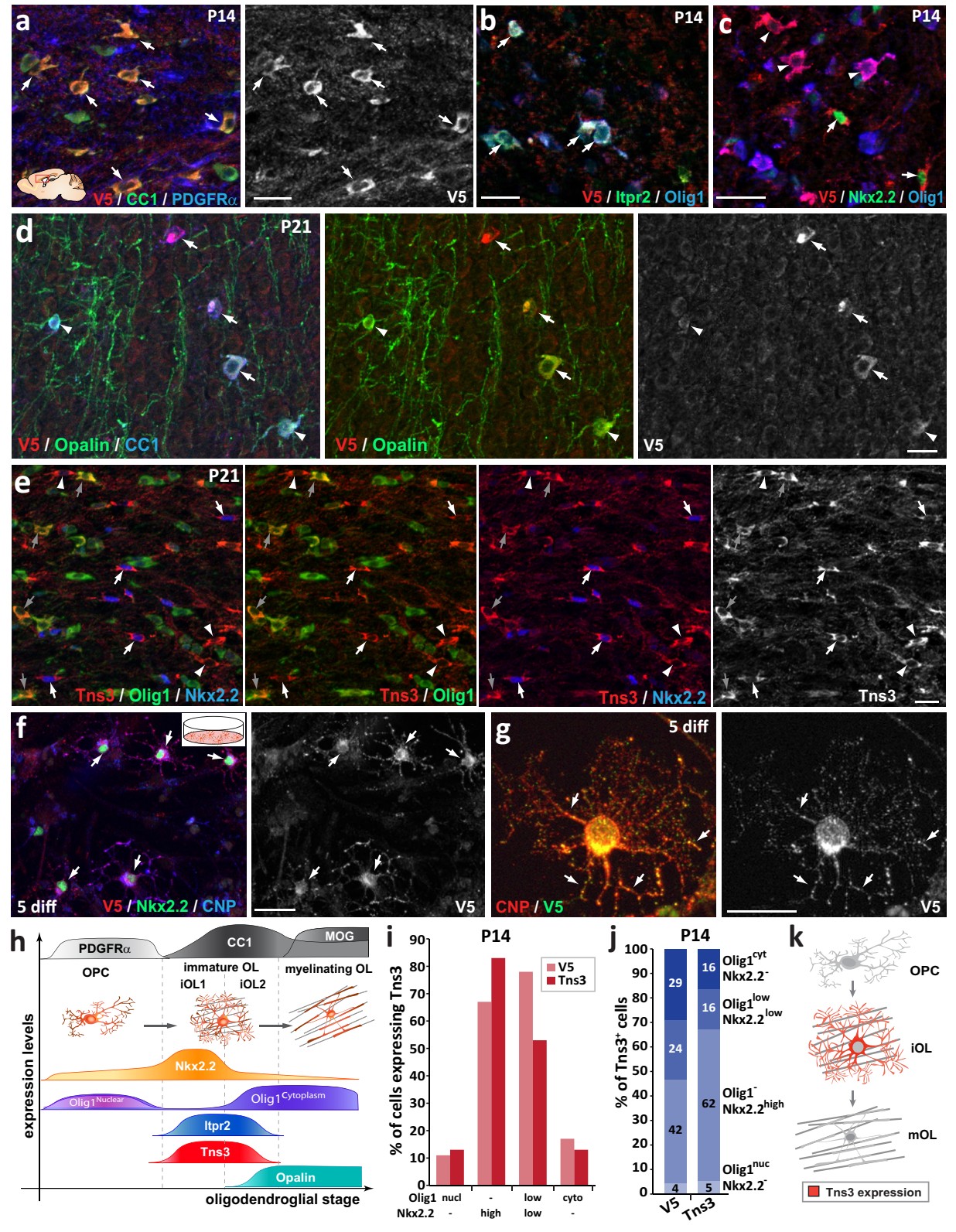

**Figure 2.** Tns3 protein is detected at high levels in the cytoplasm and main cell processes of immature oligodendrocytes (iOLs) in the postnatal brain. Immunofluorescence in sagittal sections of postnatal brain at the level of the corpus callosum at P14 (**a–c**) and P21 (**d-e**) using V5 and Tns3 antibodies. (**a**) Tns3-V5 is detected at high levels in CC1+ OLs (arrows) but not in PDGFRα+ oligodendrocyte precursor cells (OPCs). (**b**) Tns3-V5 expression overlaps well with Itpr2 (arrows), with some of them being Olig1^high-cytoplamic cells. (**c**) Tns3-V5 overlaps with high levels of Nkx2.2 expression (arrows) and also

*Figure 2 continued on next page*

*Figure 2 continued*

with Nkx2.2$^-$/Olig1$^{high\text{-}cytoplamic}$ cells (arrowheads). (**d**) Tns3-V5 expression overlaps with Opalin in iOLs (arrows, CC1$^+$ cells with large cytoplasm) but is downregulated in Opalin$^+$ myelinating oligodendrocytes (mOLs) (arrowheads, CC1$^+$ cells with small cytoplasm and myelin segments). (**e**) Tns3 (Sigma-Ct antibody) is detected at high levels in Nkx2.2$^+$/Olig1$^-$ early iOL1s (white arrows), in late Nkx2.2$^-$/Olig1$^-$ iOL1s (white arrowheads), and in Nkx2.2$^-$/Olig1$^{high\text{-}cytoplamic}$ iOL2s (gray arrows). (**f**) Tns3-V5 expression neural stem cell (NSC) cultures after 5 days in differentiation. Note the Tns3 expression in Nkx2.2$^+$/CNP$^+$ OLs (arrows). (**g**) Subcellular localization of Tns3 expression in CNP$^+$ OLs present in the cytoplasm and in dots distributed along the cell processes, overlapping with CNP signal (arrows). (**h**) Schematic representation of Tns3 expression together with key markers of different oligodendroglial stages summarizing data shown in (**a–e**). (**i**) Histograms representing the percentage of Nkx2.2$^-$/Olig1$^{high\text{-}nuclear}$, Nkx2.2$^{high}$/Olig1$^-$, Nkx2.2$^{low}$/Olig1$^{low\text{-}cytoplamic}$ and Nkx2.2$^-$/Olig1$^{high\text{-}cytoplamic}$ cells expressing Tns3-V5 and Tns3 at P14. (**j**) Histograms representing the percentage of Tns3-V5$^+$ and Tns3$^+$ cells at P14 that are Nkx2.2$^-$/Olig1$^{high\text{-}nuclear}$, Nkx2.2$^{high}$/Olig1$^-$, Nkx2.2$^{low}$/Olig1$^{low\text{-}cytoplamic}$ and Nkx2.2$^-$/Olig1$^{high\text{-}cytoplamic}$. (**k**) Schematic representation of Tns3 expression and subcellular localization in oligodendroglia. Scale bars: (**a–f**) 20 μm; (**g**) 10 μm.

The online version of this article includes the following source data and figure supplement(s) for figure 2:

**Figure supplement 1.** Immunodetection of Tns3 protein in immature oligodendrocytes (OLs) of the postnatal brain.

**Figure supplement 2.** Generation of *Tns3$^{Tns3\text{-}V5}$* knock-in mice.

**Figure supplement 2—source data 1.** Anti-V5 Western blot followed by anti-actinWestern blot at P7, P14 and P21.

**Figure supplement 3.** Tns1 and Tns2 proteins are detected at low levels in immature oligodendrocytes (OLs).

**Figure supplement 4.** Tns3 is expressed in newly formed oligodendrocytes (OLs) during adult brain remyelination.

org/home/gene/TNS3). Indeed, we could detect both the full-length (1450 aa, 155 kDa) and the Tns3 short (C-term, 61 kDa) isoforms in O4$^+$ cells from brains at P7 and P14 stages having many iOLs, but not at P21 having mainly mOLs (*Figure 2—figure supplement 2g and h*), thus validating the specificity of the anti-V5 antibodies in recognizing Tns3 protein. We eventually found a Tns3 antibody also recognizing the C-terminal of Tns3 protein (Sigma Ct) that upon optimized immunofluorescence labeling confirmed the Tns3 expression pattern seen with the V5 antibodies. In combination with Nkx2.2 and Olig1 immunofluorescence, it showed that Tns3 is strongly detected in the cytoplasm and main cellular processes of all iOL1s, defined as Nkx2.2$^{high}$/Olig1$^-$ cells having a round nucleus and small cytoplasm (*Figure 2e, i and j*, white arrows), and it divided iOL2s, defined as Nkx2.2$^-$/CC1$^{high}$ cells, into three stages: (1) Tns3$^{high}$/Nkx2.2$^-$/Olig1$^-$ (*Figure 2e, i and j*, arrowheads), (2) Tns3$^{high}$/Nkx2.2$^-$/Olig1$^{high\text{-}cytoplamic}$ (*Figure 2e, i and j*, gray arrows), and (3) Tns3$^-$/Nkx2.2$^-$/ Olig1$^{high\text{-}cytoplamic}$ (*Figure 2e, i and j*). A similar Tns3 expression pattern and localization was found in vitro using neonatal neural progenitors' differentiating cultures, where Tns3 was detected together with CNP myelin protein in the cytoplasm and cell processes of Nkx2.2$^{high}$/CNP$^+$ differentiating OLs (*Figure 2f and g*). Altogether, these results indicate that high but transient levels of Tns3 protein characterize early iOLs (iOL1s and early iOL2s), being localized at their cytoplasm and cell processes (*Figure 2h and k*).

Finally, we investigated whether other Tensin family members were expressed in oligodendroglia, finding that Tns1 and Tns2 but not Tns4 were detectable at low levels in iOLs by immunofluorescence (*Figure 2—figure supplement 3a and b*), paralleling their low transcription levels compared to *Tns3* (*Figure 2—figure supplement 3c*; brainrnaseq.org). Therefore, Tns3 appears to be the main Tensin expressed during OL differentiation, suggesting that Tns3 function in iOLs is likely to be evolutionarily selected, and thus of biological importance in oligodendrogenesis.

## Tns3 expression is found in immature oligodendrocytes during remyelination

Given the strong Tns3 expression in iOLs during postnatal myelination, we hypothesized that Tns3 expression could be enriched during remyelination in newly formed OLs contributing to remyelination. To test this hypothesis, we performed lysolecithin (LPC) focal demyelinating lesions in the corpus callosum of adult (P90) *Tns3$^{Tns3\text{-}V5}$* and wild-type mice, and assessed for Tns3 expression at the peak of OL differentiation (8 days post-lesion) in this remyelinating model (*Nait-Oumesmar et al., 1999*). We found that while non-lesioned adult brain regions contained only sparse Tns3$^+$ iOLs (CC1$^{high}$/Olig1$^{cyto\text{-}high}$ cells), remarkably many Tns3$^+$ iOLs were detected in the remyelinating area using both V5 (*Figure 2—figure supplement 4a and c*, arrows) and Tns3 antibodies (*Figure 2—figure supplement 4b*, arrows). Quantification of Tns3$^+$ cells showed a clear increase in Tns3$^+$ iOLs around the lesion borders compared to the corpus callosum far from the lesion area (*Figure 2—figure supplement 4d*), suggesting that Tns3 expression may be a useful marker of ongoing remyelination and lesion repair.

Of note, we could also detect Tns3 expression in some microglia/macrophages in the lesion area using a combination of F4/80 antibodies (*Figure 2—figure supplement 4c*, arrowheads). Altogether, all these data indicate that Tns3 expression peaks at the onset of OL differentiation, labeling iOLs during both myelination and remyelination.

## In vivo CRISPR-mediated *Tns3* loss of function in neonatal neural stem cells impairs oligodendrocyte differentiation

To explore the role of Tns3 in OL differentiation, we first utilized a *Tns3* gene trap mouse line (*Tns3*$^{βgeo}$; *Chiang et al., 2005*) and two CRISPR-mediated indel mutation mice presumptively leading to Tns3 constitutive knockout. Analyses of these three mouse lines (*Figure 3—figure supplement 1*; see 'Materials and methods' for details) showed both developmental lethality (in line with loss-of-function variants of TNS3 causing ~80% developmental mortality in the human population; https://gnomad.broadinstitute.org; *Figure 3—figure supplement 2*; 'Materials and methods') and possible genetic compensation in Tns3 expression, making them inappropriate tools to study Tns3 function in oligodendrogenesis.

Given the tendency of cells to escape the *Tns3* loss of function upon constitutive knockout mutations, we decided to assess *Tns3* requirement during postnatal oligodendrogenesis by inducing in vivo acute *Tns3* deletion in few neural stem cells (NSCs) of the neonatal brain and tracing their cell progeny with a GFP reporter. For this, we combined the postnatal electroporation technique with CRISPR/Cas9 technology. First, we used our previously validated gRNAs targeting *Tns3* at the first coding ATG (exon 6; *Figure 3—figure supplement 3*; 'Materials and methods') inserting them in an integrative CRISPR/Cas9 plasmid also expressing the GFP reporter (*Figure 3a*), to transfect neonatal NSCs of the dorsal subventricular zone (SVZ), which generate a large number of oligodendroglial cells during the first postnatal weeks (*Kessaris et al., 2006*; *Nakatani et al., 2013*), and focused our study on glial cells by quantifying the GFP⁺ progeny of targeted NSCs, outside the SVZ and located in the dorsal telencephalon 3 weeks later (P22, *Figure 3b*). The fate of GFP⁺ cells was determined by immunodetection of GFP and glial subtype markers (CC1$^{high}$ for OLs, PDGFRα for OPCs, and CC1$^{low}$ and their unique branched morphology for astrocytes). Remarkably, brains electroporated with the CRISPR plasmids targeting *Tns3* had a twofold reduction in GFP⁺ OLs compared to brains electroporated with control plasmids, while GFP⁺ OPCs were found in similar proportions (*Figure 3c, c' and d*). The proportion of GFP⁺ astrocytes was increased by 1.5-fold, likely as a result of the large reduction in OLs, as the number of GFP⁺ astrocytes was not changed (61.3 ± 10.9 in experimental versus 57.2 ± 11.8 in controls; *Figure 3c, c' and d*). To assess whether the reduction in OLs from *Tns3*-deleted NSCs was the consequence of a reduction in OPCs generated, we assessed for possible changes in numbers, proliferation, and survival of OPC at P11, when most cortical OPCs have not yet started differentiation. We found no differences in the proportion of GFP cells being OPCs (*Figure 3e*), nor the proliferative status of GFP⁺ OPCs (MCM2⁺/PDGFRα⁺ cells; *Figure 3f*) between experimental and control brains, while the reduction of OLs was already marked (*Figure 3e*), indicating that loss of *Tns3* only affected the process of OPC differentiation into OLs.

Given the expression of two Tns3 isoforms in the brain (*Figure 2—figure supplement 2f and g*), we asked whether a deletion of both isoforms would have a greater impact in OL differentiation. We thus used two gRNAs efficiently cutting the beginning and the end of *Tns3* coding sequence (5′–3′gRNAs, 'Materials and methods') to delete the whole *Tns3* locus. We found a similar reduction of OLs in the loss of the two *Tns3* isoforms than in mutations affecting only full-length Tns3 (*Figure 3h, h' and i*), suggesting that the small Tns3 isoform does not play an additional role with full-length Tns3 in OL formation. Altogether, these results indicate that *Tns3* loss-of-function mutations in neonatal SVZ-NSCs impair OPC differentiation without apparent changes in OPC generation and proliferation, thus suggesting that Tns3 is largely required for OPC differentiation into OLs in the postnatal brain (*Figure 3g*).

## OPC-specific *Tns3* deletion impairs oligodendrocyte differentiation in the postnatal brain

Given the heterogeneity of CRISPR/Cas9-mediated indels and the difficulties to assess in vivo the penetrance of their *Tns3* loss of function, to address in more detail the consequences of *Tns3* loss of function, we designed a *Tns3* conditional knockout allele by flanking with LoxP sites the exon 9

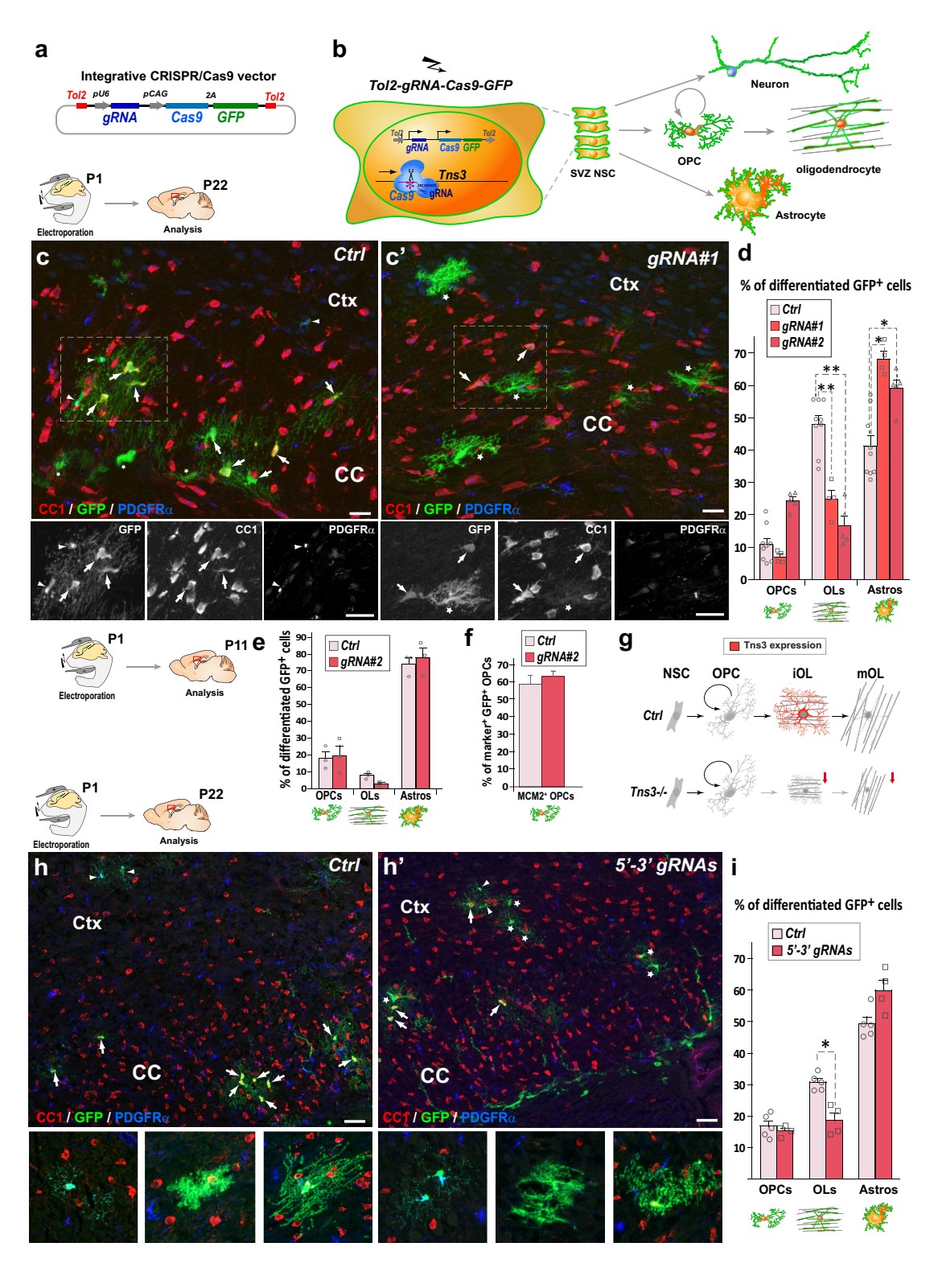

**Figure 3.** CRISPR-mediated *Tns3* mutation in neural stem cells (NSCs) reduces oligodendrocyte (OL) differentiation in the postnatal brain. (**a**) Schematic of the CRISPR/Cas9 expression vector allowing Tol2-DNA integration driving Cas9 and GFP expression (from polycistronic 2A-mediated cleaved) from CAG promoter and sgRNA expression from U6 promoter. (**b**) Schematic of the dorsal subventricular zone (SVZ) NSC electroporation of CRISPR-plasmid at postnatal day 1 (P1) and traced neural cell-subtype progeny. (**c, c'**) Immunofluorescence of representative P22 sagittal sections

*Figure 3 continued on next page*

*Figure 3 continued*

of the dorsal telencephalon showing GFP[+] cells being either PDGFRα[+] oligodendrocyte precursor cells (OPCs) (arrowheads), CC1[high] OLs (arrows), or CC1[low] astrocytes (asterisks) progeny of P1 NSCs electroporated either with *Ctrl* plasmid (**c**) or *Tns3-gRNA#1* plasmid (**c'**). (**d**) Histograms showing the percentage of GFP[+] glial cell types found in *Ctrl*, *gRNA#1* or *gRNA#2* electroporated brains being PDGFRα[+]-OPCs, CC1[high]-OLs and CC1[low]-astrocytes. Note the twofold reduction of CC1[+] OLs in *Tns3-gRNA*-transfected brains, as illustrated in (**c'**) compared to (**c**). (**e**) Histograms representing the percentage of GFP[+] differentiated cells at P11. Note the lack of changes in OPCs, and the incipient reduction in OLs. (**f**) Histograms quantifying the proportion of proliferative (MCM2[+] cells) GFP[+] OPCs in electroporated P11 mice brain. (**g**) Schematic of Tns3 expression in mice (upper) and of the effects of *Tns3* CRISPR-mediated deletion (lower). (**h, h'**) Representative P22 sagittal sections of the dorsal telencephalon showing GFP[+] cells being either PDGFRα[+] OPCs (arrowheads), CC1[high] OLs (arrows), or CC1[low] astrocytes (asterisks) progeny of P1 NSCs electroporated either with *Ctrl* plasmid (**h**) or *Tns3-5'–3'* targeting plasmid (labeled as *5'–3' gRNAs*) (**h'**). (**i**) Histograms showing the percentage of GFP[+] glial cell types found in *Ctrl* or *3'–5' gRNA* electroporated brains being PDGFRα[+]-OPCs, CC1[high] OLs, and CC1[low] astrocytes in the corpus callosum (CC) and cortex (Ctx). Note the twofold reduction of CC1[+] OLs in *Tns3-5'–3' gRNA*-transfected brains. Scale bar, 20 µm.

The online version of this article includes the following source data and figure supplement(s) for figure 3:

**Source data 1.** Subcloning strategy of to generate the Tol2-pCAG-Cas9-2A-GFP plasmid.

**Figure supplement 1.** Oligodendrogenesis is normal in *Tns3* constitutive mutant mice, which still express *Tns3* full-length transcripts in the brain.

**Figure supplement 2.** Intolerance for *TNS3* loss-of-function variants in the human population.

**Figure supplement 3.** Generation and validation of *Tns3*-targeting CRISPR/Cas9 tools.

**Figure supplement 3—source data 1.** Polyacrylamine gel (PAGE) showing efficient cutting of Tns3 gRNAs #1 and #2 as extrabands formed by hybrid DNA pairing of indels and wild-type PCR products of Tns3 targeted region.

**Figure supplement 3—source data 2.** gRNA selection strategy, targeting either the ATG region of Tns3 locus (5') gRNA#1 and gRNA#2, and the stop codon (3') region of Tns3, gRNA#3.

(*Figure 4—figure supplement 1*). In order to specifically delete *Tns3* in postnatal OPCs, we administered tamoxifen at P7 to *Pdgfra-CreER[T]; Tns3[fl/fl]; Rosa26[stop-YFP]* (hereafter called *Tns3-iKO* mice) and control pups (*Pdgfra-CreER[T]; Tns3[+/+]; Rosa26[stop-YFP]* littermates) and analyzed its effects on oligodendrogenesis at P14 and P21 (*Figure 4a*) both in white matter (corpus callosum and fimbria) and gray matter regions (cortex and striatum). We first assessed for the efficiency of *Tns3* deletion in Nkx2.2[+]/GFP[+] iOLs from different regions by immunofluorescence using a Tns3 antibody (Sigma Ct), finding that the strong Tns3 signal present in Nkx2.2[+]/GFP[+] iOLs of control brains was almost completely eliminated in *Tns3-iKO* iOLs without affecting Tns3 expression in vessels (*Figure 4—figure supplement 2b,b',c*, arrows and arrowheads versus asterisks). We then assessed for changes in oligodendrogenesis. Remarkably, the number of OLs (CC1[+]/GFP[+] cells) was reduced by half in all quantified regions (reduction of 38.95% in the CC, 48.60% in cortex, 50.88% in the fimbria, 38% in the striatum; *Figure 4b–d, Figure 4—figure supplement 2e–f', Figure 4—figure supplement 3b and b'*) in *Tns3-iKO* compared to control, while OPC (PDGFRα[+]/GFP[+] cells) density was unchanged (*Figure 4b–d*). Using markers distinguishing different stages of OL differentiation (iOL1 and iOL2/mOL), we found that the density of iOL1s (Nkx2.2[high] cells), which express the highest levels of Tns3 protein in control brains, was unchanged (*Figure 4—figure supplement 2b,b',d*), while the density of early iOL2s (CC1[+]/Olig1[-] cells) and later OL stages (iOL2/mOLs CC1[+]/Olig1[+] cells) was reduced by 30 and 50%, respectively, in *Tns3-iKO* compared to controls (*Figure 4e–f*), suggesting that Tns3 is required for normal OL differentiation. Finally, we assessed possible changes in OPC proliferation by immunodetection of Mcm2, finding no significant changes in the proliferation of *Tns3-iKO* OPCs compared to control OPCs (*Figure 4—figure supplement 3c,c',d*).

Altogether, these results indicate that acute deletion of *Tns3* in OPCs reduces by twofold generation of OLs in the postnatal brain, without major changes in OPC numbers and proliferation (*Figure 4g*).

### *Tns3-iKO* oligodendroglia undergo apoptosis

Tensins are known to mediate integrin stabilization and activation in other cell types (*Liao and Lo, 2021*), with Tns3 been shown to bind integrin-β1 through its phosphotyrosine-binding domain and FAK through its SH2 domain in fibroblasts (*Cui et al., 2004; Liao et al., 2007; Georgiadou et al., 2017*). In oligodendroglia, integrin α6β1 association with Fyn kinase is required to amplify PDGF survival signaling and promote myelin membrane formation by switching neuregulin signaling from a PI3K to a MAPK pathway (*Colognato et al., 2004*). Moreover, by conditional ablation of integrin-β1 in vivo, it was demonstrated that integrin-β1 signaling is involved in survival of differentiating oligodendroglia, but not required for axon ensheathment and myelination per se (*Benninger et al.,*

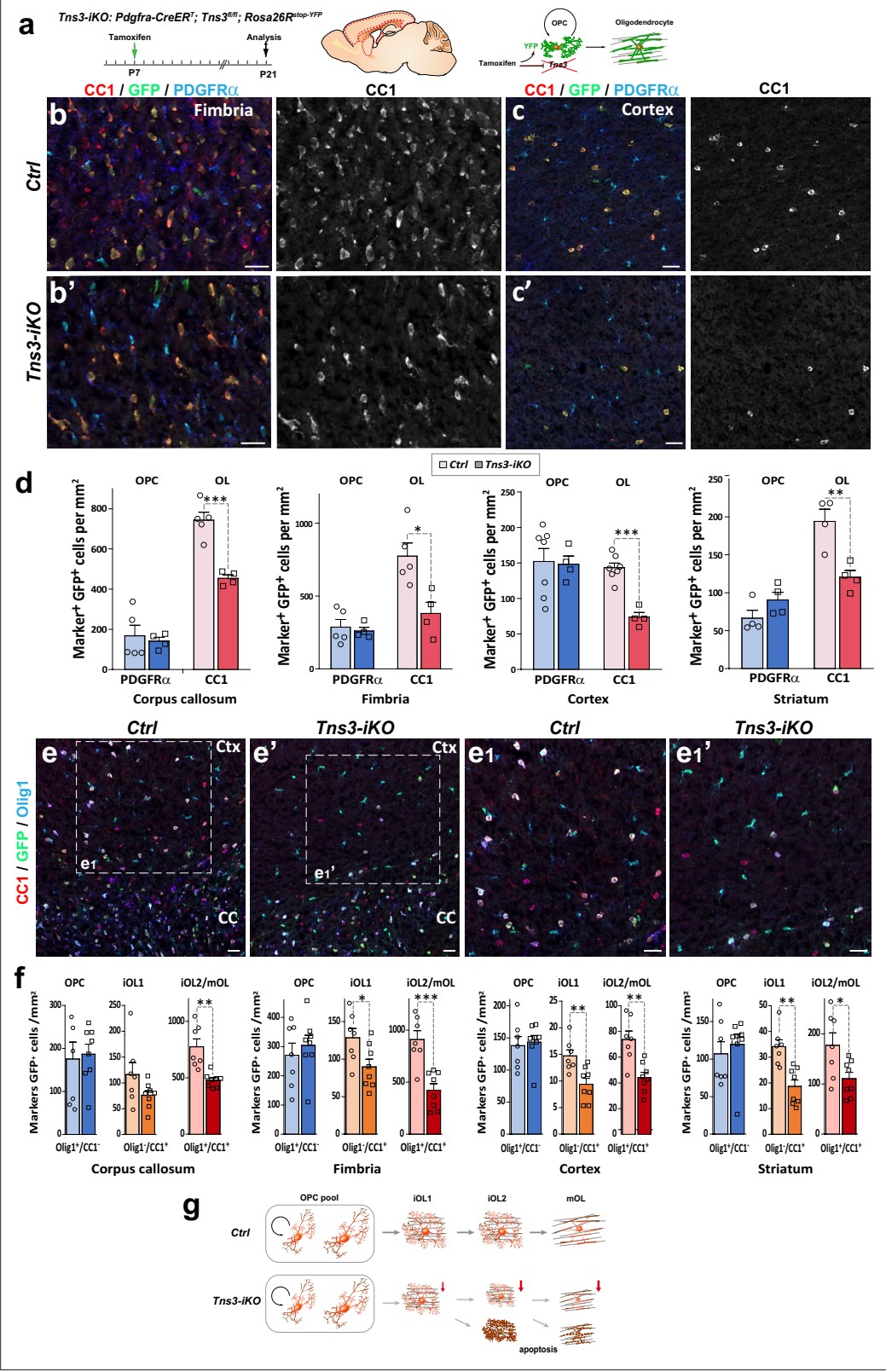

**Figure 4.** Oligodendrocyte precursor cell (OPC)-specific *Tns3* deletion reduces the number of differentiating oligodendrocytes (OLs) in the postnatal brain. (**a**) Scheme of tamoxifen administration to *Tns3-iKO* and control (*Cre⁺; Tns3⁺/⁺*) mice, Cre-mediated genetic changes, and timing of experimental analysis. (**b, b', c, c'**) Immunofluorescence in P21 sagittal brain sections for CC1, GFP, and PDGFRα illustrating similar density of OPCs

*Figure 4 continued on next page*

*Figure 4 continued*

(PDGFRα⁺) and twofold reduction in OL (CC1⁺) density in *Tns3-iKO* (**b', c'**) compared to control (**b, c**) in the fimbria (**b**) and the cortex (Ctx) (**c**). (**d**) Histograms showing OPC and OL density in P21 *Tns3-iKO* and control (*Ctrl*) mice, in the corpus callosum (CC), fimbria, Ctx, and striatum. Note the systematic OL decrease of 40–50% in each region. (**e–e₁'**) Immunofluorescence in P21 sagittal brain sections for Olig1, GFP, and CC1 to distinguish three stages of oligodendrogenesis: OPCs (Olig1⁺/CC1⁻), iOL1s (CC1⁺/Olig1⁻), and iOL2s/mOLs (CC1⁺/Olig1⁺) in *Ctrl* (**e**) or *Tns3-iKO* mice (**e'**). (**e₁**) and (**e₁'**) are higher magnification of the squared area in (**e**) and (**e'**). (**f**) Histograms showing the OPCs, the iOL1s, and the iOL2s/mOLs density in P21 *Tns3-iKO* and control mice, in the CC, fimbria, Ctx, and striatum. Note the decrease of iOL1s and iOL2s over 40% in each area quantified (except for iOL1 density in the CC). (**g**) Schematic representing defects in oligodendrogenesis found in *Tns3-iKO* compared to control. Scale bar, 20 μm.

The online version of this article includes the following figure supplement(s) for figure 4:

**Figure supplement 1.** Generation of *Tns3* floxed allele.

**Figure supplement 2.** Efficient oligodendrocyte precursor cell (OPC)-specific *Tns3* deletion in the postnatal brain and reduced oligodendrocyte generation from OPCs.

**Figure supplement 3.** Oligodendrocyte precursor cell (OPC)-specific *Tns3* deletion reduces oligodendrocyte (OL) generation without changing OPC proliferation in the postnatal brain.

*2006*). We, therefore, investigated the expression of genes involved in integrin signaling in the transcriptome of oligodendroglial cells. Indeed, *Tns3* expression pattern in iOLs was closely matching that of *Itgb1* (integrin-b1), *Fyn*, *Bcar1/p130Cas*, and *Ptk2/Fak* both in mouse and human oligodendroglia (*Figure 5—figure supplement 1a and b*). Furthermore, using neural progenitor differentiation cultures, we observed co-expression of integrin-β1 and Tns3 in CNP⁺ OLs by immunofluorescence (*Figure 5—figure supplement 1c*), suggesting that Tns3 could relay integrin-β1-mediated survival signal in differentiating oligodendroglia. Therefore, we assessed for signs of cell death in *Tns3-iKO* oligodendroglia by performing the TUNEL technique together with GFP and CC1 immunodetection. Interestingly, we found a fivefold increase in TUNEL⁺ cells in the dorsal telencephalon of *Tns3-iKO* brain, compared to control, without significant changes in non-oligodendroglial cells present in the SVZ (*Figure 5a–c*). To gain more insight into the cellular alterations and cell death of *Tns3*-deleted oligodendroglia, we investigated their cellular morphology and behavior by video microscopy during their differentiation in culture. To this end, we MACS-purified OPCs from *Tns3-iKO* and control (*Tns3^flox/flox*; *Rosa26^stop-YFP* littermates) mice at P7, 2 days after administration of tamoxifen, plated them in proliferating medium for 3 days, and recorded their behavior during 3 days in the presence of differentiation medium (*Figure 5d*). Using the expression of the YFP as a readout of Cre-mediated recombination, we compared the behavior of YFP⁺ cells (*Tns3-iKO*) with neighboring YFP⁻ cells (internal control) in the same cultures. In parallel, we used MACSorted cells from control mice as external control. Quantification of the proportion of YFP⁺ cells over time showed a 20% reduction of YFP⁺ cells (from 80% to 60%) during the 3 days in proliferation medium followed by a reduction to 50% by day 3 in differentiation medium (*Figure 5e*), suggesting possible cell death of *Tns3*-mutant cells. Live-imaging monitoring of cell behavior showed that once YFP⁺ cells had developed multiple branched morphology, characteristic of differentiating OLs, they showed a fourfold increase in their probability to die compared to YFP⁻ cells of the same culture (*Figure 5f–h*, yellow and white arrows, respectively) or to cells from control cultures, with more pronounced cell death by the third day of culture (*Figure 5f and g*). Together, these results indicate that *Tns3-iKO* oligodendroglia present increased cell death both in vivo and in primary cultures at the stage when Tns3 is upregulated and cells start to develop their branched morphology, suggesting that Tns3 likely mediates β1-integrin signaling required for their survival.

## Apoptosis of *Tns3-iKO* oligodendroglia is mediated by p53 upregulation

To study the molecular mechanisms of Tns3 function in oligodendroglia, we first looked at p53 expression, the master transcriptional regulator of the cellular genotoxic stress response (*Kastenhuber and Lowe, 2017*; *Aubrey et al., 2018*). Interestingly, we found a tenfold increase in p53⁺ OPCs (GFP⁺/CC1⁻ cells) and fourfold increase in p53⁺ iOLs (GFP⁺/CC1⁺ cells) in *Tns3-iKO* compared to control (*Figure 6a–c*), suggesting that the loss of Tns3 leads to an upregulation of p53, which, together with

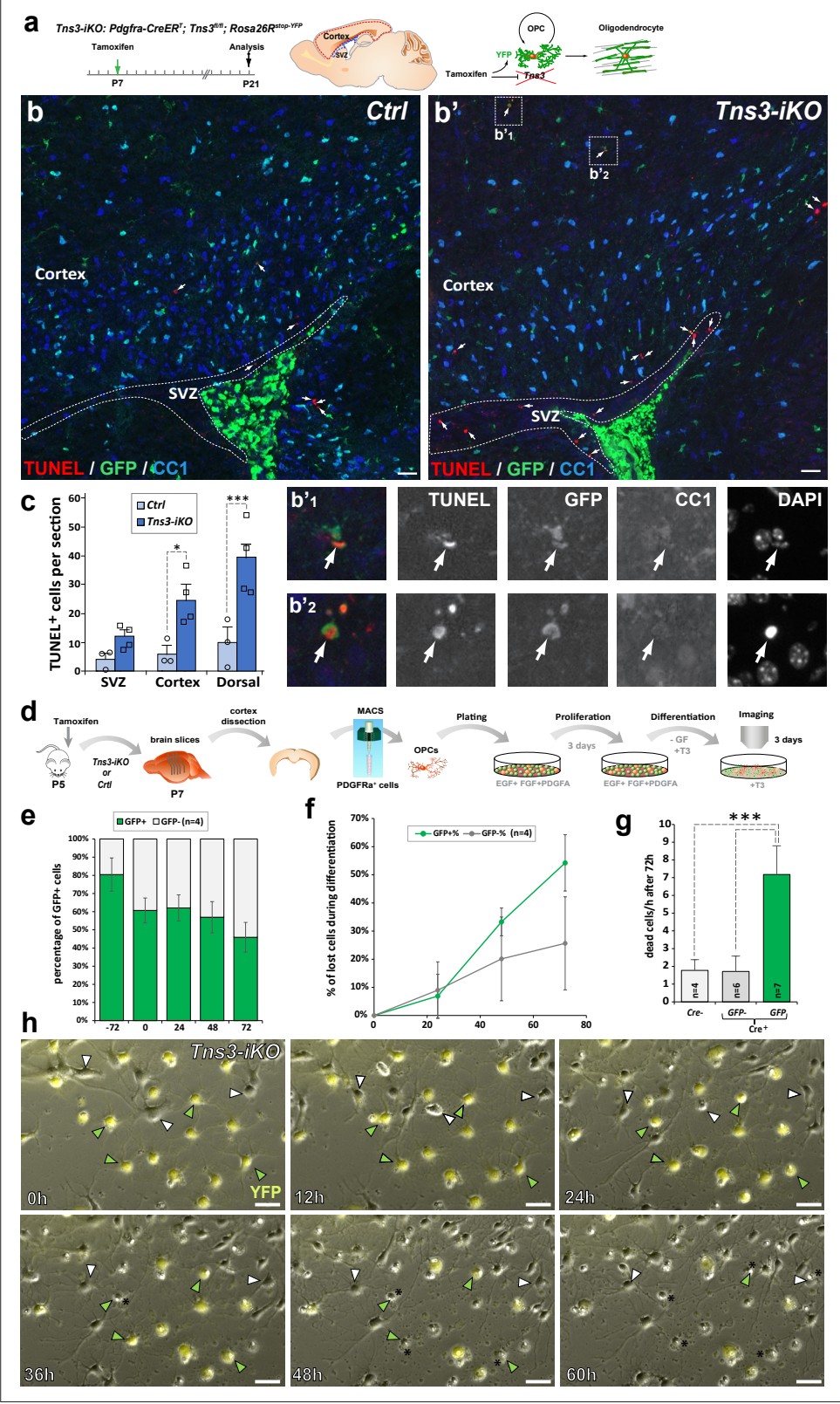

**Figure 5.** Increased cell death of *Tns3-iKO* oligodendroglia. (**a**) Scheme of tamoxifen administration to *Tns3-iKO* (*Cre+; Tns3fl/fl*) and control (*Cre+; Tns3+/+*) mice, Cre-mediated genetic changes, and timing of experimental analysis. (**a–b′**) Immunodetection of death cells by TUNEL together with recombined cells (GFP) and oligodendrocytes (OLs) (CC1+ cells) showing increased number of TUNEL+ cells in the cortex and corpus callosum of *Tns3-iKO* mice

*Figure 5 continued on next page*

*Figure 5 continued*

(**b′**) compared to control mice (**b**), including some GFP⁺/CC1⁻ oligodendrocyte precursor cells (OPCs) (insets **b′₁** and **b′₂**). (**c**) Histograms quantifying the number of TUNEL⁺ cells in the subventricular zone (SVZ), cortex, and both together (dorsal) per section. (**d**) Scheme of the video microscopy protocol in MACSorted OPCs purified from *Tns3-iKO* and control P7 mice. Cells were imaged every 10 min during 72 hr in differentiation medium. (**e**) Histograms showing the reduction of the GFP⁺ cells proportion from the plating (–72 hr) to the end of the experiment (72 hr after differentiation onset). (**f**) Time curve quantifying the loss of GFP⁺ OLs compared to GFP⁻ OLs during the 72 hr of differentiation. (**g**) Histograms representing the quantification of cells lost per hour during the 72 hr differentiation period, showing a fivefold increase in loss of GFP⁺ *Tns3-iKO* cells compared to GFP⁻ cells (nonrecombined cells from *Tns3-iKO* mice, internal negative control) or cells coming from *Cre⁻* littermates (*Cre⁻*, external negative control). (**h**) Time-lapse frames showing cells every 12 hr illustrating both GFP⁺ (green arrowheads, recombined *Tns3-iKO* cells) and GFP⁻ (white arrowheads, nonrecombined *Tns3-iKO* cells) that die over the time of video microscopy. Note the larger number of GFP⁺ OLs (cells with multibranched OL morphology) dying compared to GFP⁻ OLs. Scale bar, 20 μm.

The online version of this article includes the following figure supplement(s) for figure 5:

**Figure supplement 1.** *Tns3* is co-expressed with integrin signaling genes in immature oligodendrocytes (iOLs).

---

the loss of integrin-β1 survival signal, mediates the cell death of *Tns3-iKO* differentiating oligodendroglia (*Figure 6d*).

## *Tns3-iKO* oligodendroglia shows transcriptional dysregulation of genes involved in OPC differentiation, apoptosis, integrin signaling, and cell cycle regulation

To further unravel the defects of *Tns3*-deleted oligodendroglia, we purified oligodendroglia (O4⁺ cells) from P12 *Tns3-iKO* and control cortices by MACS (*Figure 7a*). Upon validation of Tns3 deletion at the transcript and protein levels (*Figure 7—figure supplement 1a–c*) and that similar proportion of oligodendroglia were present in each genotype (*Figure 7—figure supplement 1d–f*), we compared their transcriptomes obtained by bulk RNA sequencing (*Figure 7a*). Principal component analysis of Tns3-iKO and control samples shows clear separation between the groups (*Figure 7—figure supplement 1g*). Statistical analyses using edgeR (*Chen et al., 2016*) showed 2082 differentially expressed genes (DEGs, p-value<0.05) between *Tns3-iKO* and control, with 834 downregulated and 1248 upregulated genes (*Figure 7b*). Gene Ontology (GO) analysis of biological processes indicated that main downregulated processes were involved in terms related OL differentiation (including gliogenesis, glial cell differentiation, OL differentiation, lipid metabolism, and positive regulation of cell projection organization; *Figure 7c and e*, *Supplementary file 1*), while the upregulated biological processes related to terms such as cellular stress and p53 pathway (including double-strand break repair, cellular response to oxidative stress, and signal transduction of p53 class mediator), opposite processes involved in cell cycle regulation (including DNA integrity check point, G2/M transition of mitotic cell cycle, and positive- and negative regulation of cell cycle process) (*Figure 7d, f and g*, *Supplementary file 1*). Interestingly, GO processes related to integrin signaling and cell adhesion were upregulated (including positive regulation of cell adhesion, regulation of cell adhesion mediated by integrin, integrin-mediated signaling pathway, and positive regulation of cell adhesion; *Figure 7d*, *Supplementary file 1*), with several integrin transcripts upregulated (*Itgam, Itga8, Itgb2, Itga4, Itgb3, Itgb5, Itga6, Itgb8*, which are normally not expressed in OPCs/iOLs; *Supplementary file 1*), while *Fyn*, Src family kinase that associates with α6β1 and is required to amplify PDGF survival signaling (*Colognato et al., 2004*) was downregulated (1.4-fold, p-value=0.03; *Supplementary file 1*), suggesting that Tns3 deletion impairs normal integrin signaling, and as a consequence *Tns3*-deleted cells try to compensate this impairment upregulating of other integrin family members. These results, confirming and expanding those obtained by immunofluorescence analyses, led us to propose a model suggesting that *Tns3*-deleted oligodendroglia present signs of cellular stress accompanied by double-strand break signaling upregulation (including ATM and CHK2 regulators), p53 stabilization and upregulation of p53 target genes involved in apoptosis (including PUMA, APAF1, and Caspase 7; *Figure 7h*), and conflicting signals related to cell cycle (upregulation of p21, promoting cell cycle arrest, and upregulation of CDK/cyclin complexes promoting cell cycle progression; *Figure 7h*).

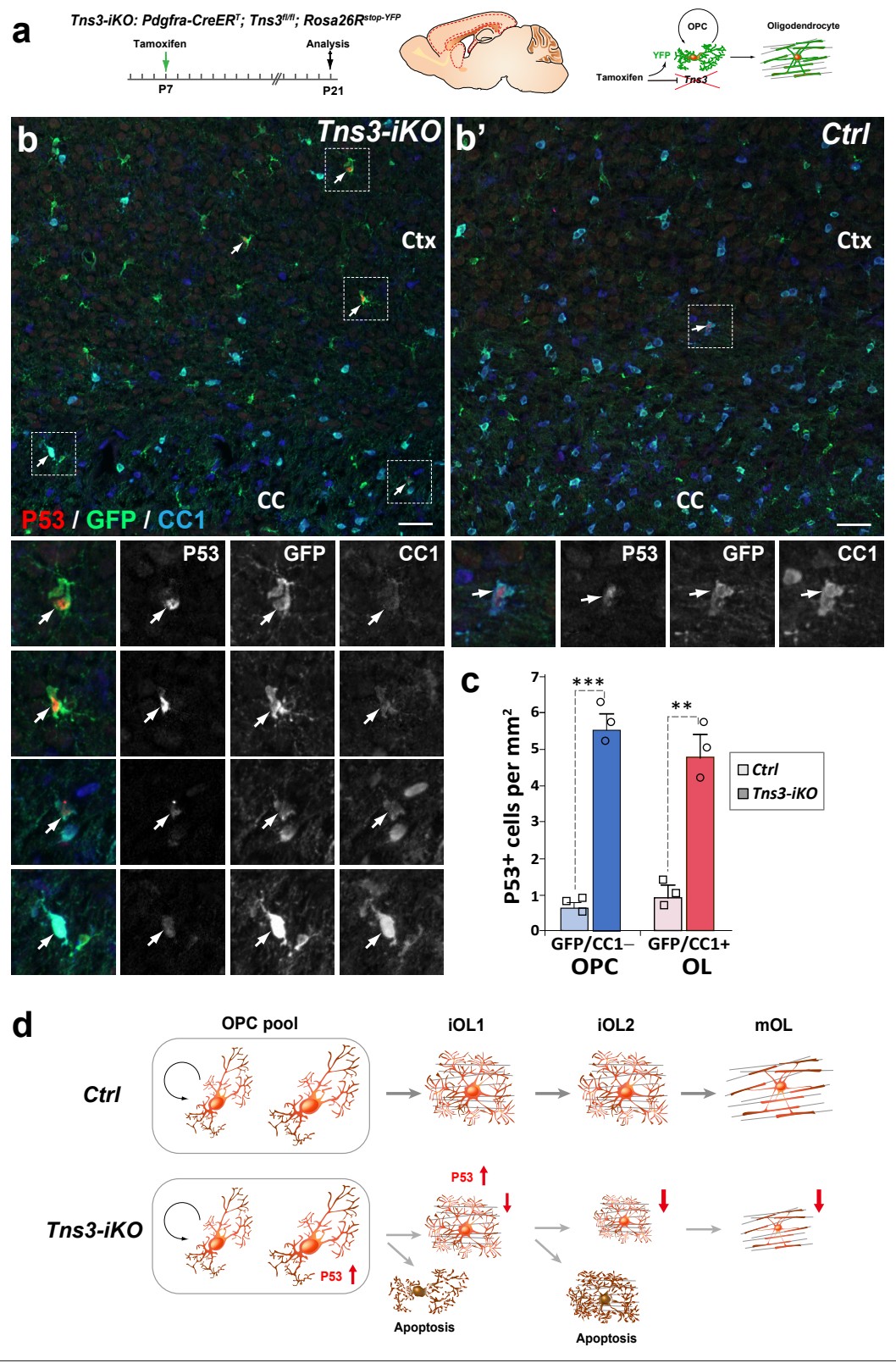

**Figure 6.** p53-mediated cell death of *Tns3-iKO* oligodendroglia. (**a**) Scheme of tamoxifen administration to *Tns3-iKO* (*Cre⁺; Tns3^fl/fl*) and control (*Cre⁺; Tns3^+/+*) mice, Cre-mediated genetic changes, and timing of experimental analysis. (**b, b'**) Immunodetection of p53 together with GFP to label recombined cells and CC1 to label oligodendrocytes (OLs) showing a strong increased number of p53⁺/GFP⁺/CC1⁻ oligodendrocyte precursor cells

*Figure 6 continued on next page*

*Figure 6 continued*

(OPCs) and p53⁺/GFP⁺/CC1⁺ OLs in the cortex (Ctx) and corpus callosum (CC) of *Tns3-iKO* mice (**b**) compared to control mice (**b′**). Dotted squares highlight some cases of p53⁺ cells, shown at higher magnification below. (**c**) Histograms quantifying the number of p53⁺ cells per area (mm²) in the dorsal telencephalon. Scale bar, 20 μm. (**d**) Schematics representing *Tns3*-deleted phenotypes in oligodendroglia. Scale bar, 20 μm.

## Discussion

The tight balance of OPCs between proliferation, survival, and differentiation ensures their capacity to respond to the myelination needs of the CNS by generating new OLs on demand, whilst avoiding the generation of brain gliomas through uncontrolled OPC proliferation. The observation that OPCs are present within demyelinating MS lesions but fail to efficiently differentiate into myelinating cells with age and disease progression (*Chang et al., 2002*; *Neumann et al., 2019*), together with the strong sensitivity of iOLs to survival/apoptotic signals (*Hughes and Stockton, 2021*), suggests that efforts to foster OPC differentiation and survival of iOLs are a critical events for healthy aging and successful remyelination in MS patients. In this study, we combined the genome-wide binding profile of key regulators of OL differentiation, Olig2, Chd7, and Chd8 (*Lu et al., 2000*; *Zhou et al., 2000*; *Lu et al., 2002*; *Zhou and Anderson, 2002*; *He et al., 2016*; *Küspert and Wegner, 2016*; *Marie et al., 2018*; *Zhao et al., 2018*), to identify their common gene targets, and focused our analysis on Tensin3 (Tns3), whose expression matched the onset of OL differentiation. To study Tns3 expression and function, we generated several genetic tools, including CRISPR/Cas9 vectors, to induce *Tns3* mutations both in vivo and in vitro, a *Tns3^Tns3-V5* knock-in mouse, two constitutive *Tns3* knockout mice, and finally an inducible knockout (*Tns3^Flox*) mouse. Using these tools, we provide several lines of evidence showing that Tns3 is upregulated in iOLs and required for normal OL differentiation. First, we show that Tns3 expression is strongly induced at the onset of OL differentiation, localized to the cytoplasm and main cell processes of iOLs, and downregulated in mature OLs both at the transcript and protein levels, thus constituting a novel marker for iOLs, for which we provide an optimal immunofluorescence protocol with a commercial antibody (Sigma, Ct). Second, we show that during remyelination Tns3 is also expressed in newly formed OLs and thus could be used as a hallmark for ongoing remyelination. Third, analyzing both *Tns3^βgeo* gene trap mice and two *Tns3^KO* mice, we show that constitutive *Tns3* deletion is detrimental for normal development and that the predicted loss of Tns3 full-length transcript and protein is bypassed in the oligodendroglia of surviving homozygous animals, paralleling the intolerance for *TNS3* loss-of-function variants found in the human population. Fourth, in vivo CRISPR-mediated *Tns3* deletion in neonatal NSCs from the SVZs leads to a twofold reduction of OLs without changes in OPC generation, proliferation, and numbers. Fifth, in vivo Tns3-induced knockout (*Tns3-iKO*) in postnatal OPCs leads to a twofold reduction of differentiating OLs without reducing the overall OPC population, both in gray and white matter brain regions. Finally, we provide evidence, by immunodetection in vivo and video microscopy of primary OPC differentiation cultures, that *Tns3-iKO* differentiating oligodendroglia upregulate p53, key sensor of cell stress, and present a four- to five-fold increase in apoptosis compared to control oligodendroglia, suggesting that mechanistically Tns3 function is likely required for normal OL differentiation at least in part by mediating integrin-β1 survival signaling in differentiating oligodendroglia.

### Tns3 is a novel marker for immature oligodendrocytes

Recent studies have started to uncover genes enriched in iOLs, such as *Itpr2* (*Zeisel et al., 2015*; *Marques et al., 2016*), *Enpp6* (*Xiao et al., 2016*), and *Bcas1* (*Fard et al., 2017*), that could be used as markers for these transient cell populations, particularly interesting to label areas of active (re) myelination in the context of OL and myelin pathology, such as preterm brain injury and MS. Here, we report for the first time that Tns3 is a hallmark of iOLs (*Figure 2*). Tns3 is expressed at high levels in iOLs and downregulated as OLs mature into myelinating cells, showing a complete overlap with Itpr2 transcript and protein. We found that a commercial Tns3 antibody (Millipore) also recognizes another nuclear protein that, like Tns3 in the cytoplasm, also labels at high levels iOLs, paralleling the case of CC1 antibody, which recognizes both APC and Quaking-7 proteins in OLs (*Lang et al., 2013*; *Bin et al., 2016*). Upon testing several antibodies, we found one (Sigma Ct) optimally labeling iOLs by immunofluorescence in brain sections and oligodendroglial cell cultures, whereas the Itpr2 commercial antibody we tried did not match this high-quality iOLs immunolabeling. An optimized

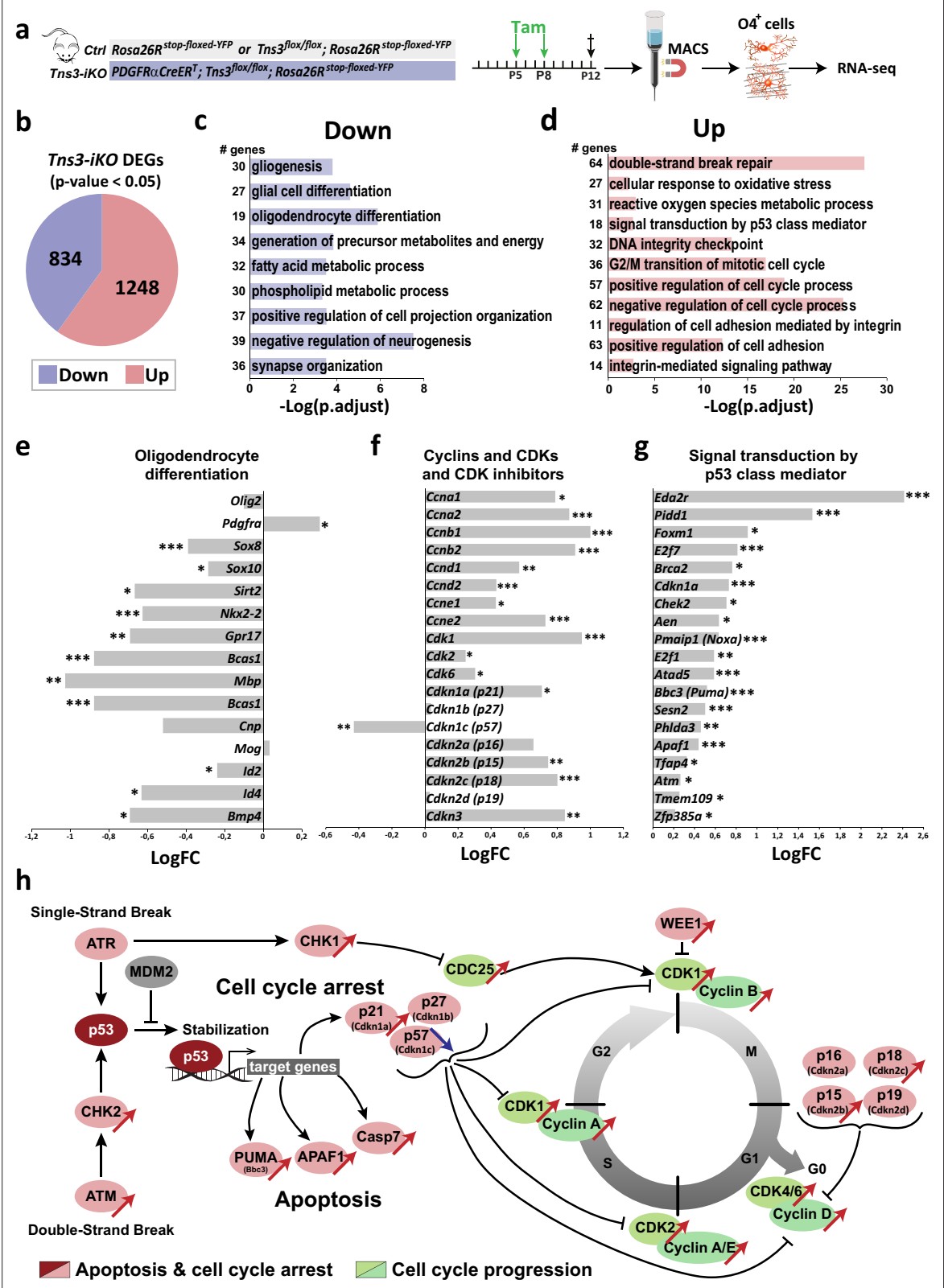

**Figure 7.** Mechanisms involved in *Tns3-iKO* oligodendroglial defects. (**a**) Diagram representing tamoxifen (Tam) injection in P5 and P8 *Ctrl* and *Tns3-iKO* pups followed by MACSorting of O4+ cells that were used to do RNA-seq. (**b**) Pie chart showing the amount of differentially expressed genes (DEGs; p-value<0.05) in *Tns3-iKO* O4+ cells compared to *Ctrl*. (**c, d**) Gene Ontology (GO) analysis of biological processes downregulated (**c**) and upregulated (**d**) in *Tns3-iKO* O4+ cells compared to *Ctrl*. Numbers on left represent the number of deregulated genes in each GO process. (**e–g**) Graphs representing

*Figure 7 continued on next page*

*Figure 7 continued*

the logarithmic fold change (LogFC) for an example of genes involved in oligodendrocyte differentiation (**e**), cell cycle (**f**), and p53 pathway (**g**). (**h**) Summary schematics of the transcriptional dysregulation of *Tns3-iKO* O4[+] cells, representing the upregulation of the apoptosis pathway and the conflicting signals on cell cycle arrest/progression. Red and blue arrows represent gene upregulation and downregulation, respectively, in *Tns3-iKO* O4[+] cells compared to *Ctrl.*

The online version of this article includes the following figure supplement(s) for figure 7:

**Figure supplement 1.** Transcriptomic analysis of acutely Tns3-deleted oligodendroglia.

protocol for immunodetection using Bcas1-recognizing antibodies has been shown to label iOLs (*Fard et al., 2017*). Finally, Enpp6 is very specific for iOLs at the transcript level (*Xiao et al., 2016*), but to our knowledge, no Enpp6-recognizing antibodies producing good quality immunodetection are yet available. Therefore, Tns3 protein expression in the CNS is a hallmark of iOLs, and the Tns3 Sigma Ct antibody is an optimal reagent to label iOLs during both myelination and remyelination.

## Tns3 is required for oligodendroglial differentiation

OL differentiation involves substantial generation of new membrane and cell processes composing the 40–60 myelin segments formed by mature OLs (*Hughes et al., 2018*). Actin cytoskeleton remodeling is an important driver of the OL morphological changes undergone during their differentiation (*Nawaz et al., 2015*; *Zuchero et al., 2015*). Tensin proteins, linking the extracellular signals received by transmembrane integrins with the actin cytoskeleton in different cell types (*Liao and Lo, 2021*), are well placed to play an important role in these morphological changes. At the molecular level, it has been shown that the phosphotyrosine-binding domains of Tensins interact with the NPXY motifs present in the cytoplasmic tails of integrin-β1 in a pTyr-insensitive fashion (*Calderwood et al., 2003*; *Katz et al., 2007*; *McCleverty et al., 2007*), allowing Tensins to bring actin filaments, through their actin binding domain, to focal adhesion sites (*Liao and Lo, 2021*). Given that the extension of OL cell processes' growth cone is guided by the sequential activation of Fyn, FAK, and RhoGAP (*Thomason et al., 2020*), and that high levels of Tns3 protein are detected in the cell body and processes of iOLs coinciding with their large enlargement, Tns3 is thus well placed to mediate integrin signaling to the actin cytoskeleton and play an active role in this large cellular remodeling. Moreover, integrin-β1, FAK/Ptk2, Fyn, p130Cas/Bcar1, and Tns3 are all highly expressed in iOLs (*Figure 5—figure supplement 1*). Here, using three independent approaches, we show that loss of Tns3 in iOLs reduces by half the numbers of OLs in the postnatal brain. It is therefore very likely that Tns3 act as a mediator of integrin α6β1 signaling to promote OL survival and differentiation by mediating actin cytoskeletal remodeling. Finally, through its additional ability to bind to EGFR (*Cui et al., 2004*), whose activation is another driver of oligodendroglial differentiation, Tns3 could also be required for mediation of signaling downstream of growth factor receptor activation in early iOLs; this could also explain the increased death in OLs lacking Tns3.

## Tns3's role in immature oligodendrocyte survival

Programmed cell death regulates developmental oligodendrogenesis, with a large proportion of iOLs degenerating before the fourth week of postnatal life in mice (*Barres et al., 1992*; *Trapp et al., 1997*). Also in the adult mouse brain, differentiating OPCs remain in the iOL stage for roughly 2 days with many of them undergoing programmed cell death (*Hughes et al., 2018*), indicating that this immature stage is very dependent on survival signals. Apoptotic pathways involving BCL-2 family members have been shown to regulate this oligodendroglial programmed cell death (reviewed in *Hughes and Stockton, 2021*). One study has shown that the transcription factor TFEB, involved in autophagy and lysosomal biogenesis, ensures the spatial and temporal specificity of developmental myelination by promoting the expression of ER stress genes and PUMA, a pro-apoptotic factor inducing Bax-Bak-dependent programmed cell death in differentiating oligodendroglia (*Sun et al., 2018*). Another recent study showed that during both during homeostasis and remyelination the activity of the primary sensor of cellular stress, nuclear factor (erythroid-derived 2)-like 2 (Nrf2), induces the expression of Gsta4, a scavenger of lipid peroxidation, which in turn controls apoptosis of iOLs via the mitochondria-associated Fas-Casp8-Bid-axis (*Carlström et al., 2020*).

Several studies have shown that integrin-β1 signaling is required for iOL survival. Neuronal-derived signals, including neuregulin and laminin-2, are received by iOLs through integrin-β1 signaling that

would enhance the function of neuroligin as a survival factor by inducing a survival-dependence switch from the phosphatidylinositol 3-kinase-Akt pathway to the mitogen-activated protein kinase (MAPK) pathway, with enhanced MAPK signaling inactivating the pro-apoptotic molecule BAD (*Colognato et al., 2002*; *Benninger et al., 2006*). Also, PDGF survival signaling in OPCs and myelin formation have been shown to depend on integrin α6β1 binding to Fyn (*Colognato et al., 2004*). Tensins typically reside at focal adhesions, which connect the extracellular matrix (ECM) to the cytoskeletal networks through integrins and their associated protein complexes (*Kumar, 1998*; *Liao and Lo, 2021*), with focal adhesions mediating both outside-in and inside-out signaling pathways that regulate cellular events, such as cell attachment, migration, proliferation, apoptosis, and differentiation (*Liao and Lo, 2021*). In this study, we show that Tns3 expression timing during oligodendrogenesis parallels that of integrin-β1, and we provided evidence of Tns3 and integrin-β1 co-localization in dotted structures resembling nascent and focal adhesion in the cytoplasm and processes of iOLs. Moreover, Tns3-deleted oligodendroglia have a 1.4-fold downregulation of *Fyn* transcripts, Src family kinase that associates with α6β1 and is required to amplify PDGF survival signaling (*Colognato et al., 2004*), accompanied by the upregulation of transcripts of other integrin family members normally not expressed in OPCs and iOLs (*Supplementary file 1*), suggesting cellular changes to compensate the integrin signaling impairment of *Tns3*-deleted cells. Remarkably, knockout mice for *integrin-α6* present a 50% reduction in brainstem MBP⁺ OLs at E18.5, just before they die at birth, accompanied by an increase in TUNEL⁺ dying OLs (*Colognato et al., 2002*), while conditional deletion of *integrin-β1* in iOLs by *Cnp-Cre* also leads to a 50% reduction in cerebellar OLs at P5, with a parallel increase in TUNEL⁺ dying OLs (*Benninger et al., 2006*). Therefore, given that *Tns3*-induced deletion in postnatal OPCs also leads to a 40–50% reduction in OLs in both gray and white matter regions of the postnatal telencephalon (this study), paralleled by a similar increase in TUNEL⁺ apoptotic oligodendroglia, we suggest that Tns3 is required for integrin-β1-mediated survival signal in iOLs. Moreover, we suggest that this would lead to cellular stress of *Tns3*-deleted differentiating oligodendroglia and to the upregulation of p53, master regulator of cellular stress and apoptosis, which has been previously been shown to be involved in the apoptosis of human OLs in the context of MS (*Ladiwala et al., 1999*; *Wosik et al., 2003*) and in the cuprizone demyelination mouse model (*Li et al., 2008*; *Luo et al., 2021*).

In summary, here we have generated powerful genetic tools allowing to assess for the first time the role of Tns3 in the CNS, showing that Tns3 protein is found at high levels in the cytoplasm and main processes of iOLs, thus constituting a new marker of this oligodendroglial stage, and demonstrated by different genetic approaches that Tns3 deletion leads to a twofold reduction in differentiating OLs, explained at least in part by their increased apoptosis due to p53 upregulation and likely the loss of integrin-β1-mediated survival signaling. Follow-up studies using these tools should unravel with more detail the molecular mechanisms mediated by Tns3 not only in iOLs during developmental myelination but also in pathological contexts such as preterm birth dysmyelination, adult demyelination in MS and glioblastoma, this last one recently associated with reduced levels of Tns3 (*Chen et al., 2017*).

## Materials and methods

**Key resources table**

| Reagent type (species) or resource | Designation | Source or reference | Identifiers | Additional information |
|---|---|---|---|---|
| Strain, strain background (*Mus musculus*) | *Tns3ᵝᵍᵉᵒ* | Su Hao Lo lab (UC Davis, USA) | | |
| Strain, strain background (*M. musculus*) | *Tn3Flox* | Our lab | | Available as collaboration |
| Strain, strain background (*M. musculus*) | *Tns3Tns3-V5* | Our lab | | Available as collaboration |
| Strain, strain background (*M. musculus*) | *Tns3KO* | Our lab | | Lost in the Covid-19 |
| Chemical compound, drug | Tamoxifen | Sigma | T5648 | |
| Chemical compound, drug | 32% PFA solution | Electron Microscopy Sciences | 50-980-495 | |

*Continued on next page*

*Continued*

| Reagent type (species) or resource | Designation | Source or reference | Identifiers | Additional information |
|---|---|---|---|---|
| Commercial assay or kit | Neural tissue dissociation kit (P) | Miltenyi Biotec | 130-093-231 | |
| Commercial assay or kit | Debris removal kit | Miltenyi Biotec | 130-090-101 | |
| Commercial assay or kit | Anti-PDGFRα coupled-beads | Miltenyi Biotec | 130-094-543 | |
| Commercial assay or kit | Anti-O4 coupled-beads | Miltenyi Biotec | 130-096-670 | |
| Chemical compound, drug | Normal goat serum | Eurobio | CAECHVOO-OU | |
| Chemical compound, drug | DAPI | Sigma-Aldrich | D9542 | |
| Chemical compound, drug | Fluoromount-G | SouthernBiotech | 15586276 | |
| Commercial assay or kit | In Situ Cell Death detection kit | Roche | 12156792910 | |
| Chemical compound, drug | RIPA buffer | Thermo Fisher | 89901 | |
| Chemical compound, drug | Halt Protease Inhibitor Cocktail | Thermo Fisher | 87786 | |
| Commercial assay or kit | Pierce Detergent Compatible Bradford Assay Kit | Thermo Fisher | 23246 | |
| Chemical compound, drug | Bolt LDS Sample Buffer | Thermo Fisher | B0007 | |
| Chemical compound, drug | 4–12% polyacrylamide gradient gels | Thermo Fisher | NW04122BOX | |
| Chemical compound, drug | Bolt MOPS SDS Running Buffer | Thermo Fisher | B0001 | |
| Chemical compound, drug | Mini Gel Tank and Blot Module Set | Thermo Fisher | NW2000 | |
| Chemical compound, drug | Precision Plus Protein All Blue protein standards | Bio-Rad | 1610373EDU | |
| Chemical compound, drug | Amersham Protran 0.2 μm nitrocellulose membrane | Dutscher | 10600001 | |
| Chemical compound, drug | NuPAGE Transfer Buffer | Thermo Fisher | NP0006-1 | |
| Chemical compound, drug | Pierce ECL Western Blotting Substrate | Thermo Fisher | 32109 | |
| Chemical compound, drug | Poly-L-ornithine | Sigma | P4957 | |
| Chemical compound, drug | DMEM/F12 | Life Technologies | 31331028 | |
| Chemical compound, drug | HEPES buffer | Life Technologies | 15630056 | |
| Chemical compound, drug | Glucose | Sigma | G8769 | |
| Chemical compound, drug | Penicillin/streptomycin | Life Technologies | 15140122 | |
| Chemical compound, drug | N2 supplement | Life Technologies | 17502048 | |
| Chemical compound, drug | B27 supplement | Life Technologies | 17504044 | |
| Chemical compound, drug | EGF | PeproTech | AF-100-15 | |
| Chemical compound, drug | FGF-basic | PeproTech | 100-18B | |
| Chemical compound, drug | PDGF-AA | PeproTech | 100-13A | |
| Chemical compound, drug | Insulin | Sigma | I6634 | |
| Chemical compound, drug | iDeal ChIPseq kit for Transcription Factors | Diagenode | C01010055 | |
| Antibody | Anti-Tensin1 (rabbit) | Su Hao Lo, UC Davis | | 1:100 |
| Antibody | Anti-Tensin2 (rabbit) | Su Hao Lo, UC Davis | | 1:100 |
| Antibody | Anti-Tensin3 (rabbit) | Sassan Hafizi, University of Portsmouth | | 1:1000 |
| Antibody | Anti-Tensin3 (rabbit) | Millipore | AB229 | 1:500 |

*Continued on next page*

*Continued*

| Reagent type (species) or resource | Designation | Source or reference | Identifiers | Additional information |
|---|---|---|---|---|
| Antibody | Anti-Tensin3 (rabbit) | Thermo Fisher | PA5-116022 | 1:1000 |
| Antibody | Anti-Tensin3 (mouse monoclonal) | Santa Cruz Biotech | sc-376367 | 1:500 |
| Antibody | Anti-Tns3 (rabbit) (C-terminal) | Sigma-Aldrich | SAB4200205 | 1:200 |
| Antibody | Anti-Tns3 (rabbit) (TN-17) | Sigma-Aldrich | SAB4200416 | 1:400 |
| Antibody | Anti-V5 tag (rabbit) | Millipore | AB3792 | 1:2000 |
| Antibody | Anti-V5 tag (mouse monoclonal) | Invitrogen | R960-25 | 1:1000 |
| Antibody | Anti-PDGFRα (rat) | BD Biosciences | 558774 | 1:250 |
| Antibody | Anti-Olig1 (mouse monoclonal) | NeuroMab | 75-180 | 1:1000 |
| Antibody | Anti-Olig2 (mouse monoclonal) | Millipore | MABN50 | 1:500 |
| Antibody | Anti-CNPase (mouse monoclonal) | Millipore | MAB326R | 1:250 |
| Antibody | Anti-IP3 receptor 2 (rabbit) (Itpr2) | Millipore | AB3000 | 1:40 |
| Antibody | Anti-Nkx2.2 (mouse polyclonal) | Developmental Studies Hybridoma Bank | | 1:4 |
| Antibody | Anti-CC1 (mouse monoclonal) (Quaking 7) | Calbiochem | OP80 | 1:100 |
| Antibody | Anti-MOG (mouse monoclonal) | ICM, Paris Hybridoma | AA3 | 1:20 |
| Antibody | Anti-Opalin (mouse monoclonal) | Santa Cruz Biotech | sc-374490 | 1:500 |
| Antibody | Anti-GFP (chicken polyclonal) | Aves Labs | GFP-1020 | 1:1000 |
| Antibody | Anti-GFP (rabbit) | Life Technologies | A6455 | 1:1000 |
| Antibody | Anti-MCM2 (mouse)(BM28) | BD Biosciences | 610701 | 1:500 |
| Antibody | Anti-p53 (rabbit) | Leica | P53-CM5P-L | 1:500 |
| Antibody | Anti-F4/80 (rat) | Abd Serotec | MCA497 | 1:100 |
| Antibody | Anti-beta-galactosidase (mouse monoclonal) | Promega | Z3783 | 1:1000 |
| Antibody | HRP-conjugated anti-rabbit | Bio-Rad | 1706515 | 1:5000 |
| Antibody | HRP-conjugated anti-mouse | Sigma-Aldrich | NA931-1ML | 1:5000 |
| Antibody | HRP-conjugated anti-rat | Thermo Fisher Scientific | A10549 | 1:5000 |
| Antibody | Anti-H3K4me3 (rabbit) | Active Motif | 39060 | |
| Antibody | Anti-H3K27Ac (rabbit) | Active Motif | 39034 | |
| Antibody | Anti-H3K27me3 (mouse) | Abcam | ab6002 | |
| Antibody | Anti-H3K4me1 (rabbit) | Ozyme | 5326T | |
| Plasmid | *gRNA-pCMV-Cas9-2A-GFP* | Addgene | | |
| Plasmid | *gRNA-pCMV-Cas9-2A-Puro* | Addgene | | |
| Plasmid | *Tol2-gRNA-pCMV-Cas9-2A-GFP* | Our lab | | |
| Plasmid | *Tol2-gRNA-pCMV-Cas9-2A-Puro* | Our lab | | |
| Plasmid | *Tol2-gRNA-pCAG-Cas9-2A-GFP* | Our lab | | |
| Plasmid | *Tol2-gRNA-pCAG-Cas9-2A-Puro* | Our lab | | |
| Plasmid | *Tol2-Tns3gRNA2-pCAG-Cas9-2A-GFP* | Our lab | | |

## Animals

All animal procedures were performed according to the guidelines and regulations of the Inserm ethical committees (authorization #A75-13-19) and animal experimentation license A75-17-72 (CP).

Both males and females were included in the study. Mice were maintained in standard conditions with food and water ad libitum in the ICM animal facilities. *Tensin3* gene trap mouse line (*Tns3^βgeo^*) was from Su Hao Lo lab (UC Davis, USA). Mice used for ChIP-seq analysis were wild-type Swiss obtained from Janvier Labs. *Tns3^flox^* were crossed with *Pdgfra-CreER^T^* (**Kang et al., 2010**) and *Rosa26^stop-floxed-YFP^* mice to generate *Tns3^flox^; Pdgfra-CreER^T^; Rosa26^stop-floxed-YFP^* mice line. *Pdgfra-CreER^T^; Rosa26^stop-floxed-YFP^* mice were used as controls.

## Generation of *Tns3^Tns3-V5^* knock-in mice

*Tns3^Tns3-V5^* mice were generated at the Curie Institute mouse facility. Briefly, the Cas9 protein, the crRNA, the tracrRNA, and an ssODN targeting vector for the *Tns3* gene had been microinjected into a mouse egg cell, which was transplanted into a C57BL/6J-BALB/cJ female surrogate. Pups presenting HDR insertion of the V5 tag were selected after genotyping.

## Generation of *Tns3^4del^* and *Tns3^14del^* knockout mice

*Tns3^KO^* mice were generated at the ICM mice facility. Briefly, the Cas9 protein, the crRNA, the tracrRNA, and a targeting vector for the *Tns3* gene had been microinjected into a mouse egg cell transplanted into a C57BL/6J female surrogate. Pups with NHEJ mutations inducing a gene frameshift were selected after genotyping and Sanger sequencing verification. Finally, only two lines containing indels of 4 and 14 nucleotide deletions were maintained and studied.

## Generation of *Tns3^Flox^* mice and tamoxifen administration

We designed a *Tns3* conditional knockout allele by flanking with LoxP sites exon 9 (LoxP-Exon9-LoxP; **Figure 4—figure supplement 1a**). In this *Tns3*-floxed allele (*Tns3^flox^*), Cre-mediated recombination induces a transcription frame shift translated into an early stop codon, leading to a putative small peptide of 109 aa instead of the full-length Tns3 protein (1442 aa; **Figure 4—figure supplement 1b and c**). *Tns3^flox^* mice were generated at the Transgenic Core Facility of the University of Copenhagen. The repair template contained homology arms of 771 bp and 759 bp length and *loxP* sequences flanking exon 9, was synthesized by Invitrogen, and verified by Sanger sequencing. Two gRNAs were designed at the Transgenic Core Facility that target a DNA sequence in the proximity of each *loxP* site. The gRNAs were designed in a fashion where the insertion of the *loxP* disrupts the targeting site, thus preventing retargeting of the repaired DNA. Mouse embryonic stem cell (mESC) method was used for the generation of this mouse model by transfecting ESCs with the repair construct (dsDNA) together with two plasmids – each containing each gRNA. Identification of the positive mESC clones was done via a combination of a PCR genotyping and Sanger sequencing confirmation. Mouse ESCs were transfected with a plasmid expressing *Cas9, GFP*, and gRNAs flanking *Tns3* exon 9, and a *Tns3*-floxed targeting vector (**Figure 4—figure supplement 1d**), in order to induce CRISPR/Cas9-mediated homologous recombination. After verifying the presence of *Tns3*-floxed allele in *Tns3* locus by Sanger sequencing, positive ESC clones were injected into blastocysts to generate *Tns3*-floxed (*Tns3^flox^*) mice.

Tamoxifen (Sigma, T5648) was dissolved in corn oil (Sigma, C-8267) and injected subcutaneously at 20 mg/ml concentration at P7 (30 µl) in *Ctrl* and *Tns3-iKO* animals. Brains were then collected at P21.

## Postnatal electroporation

Postnatal brain electroporation (**Boutin et al., 2008**) was adapted to target the dorsal SVZ. Briefly, postnatal day 1 (P1) pups were cryoanesthetized for 2 min on ice and 1.5 µl of plasmid was injected into their left ventricle using a glass capillary. Plasmids were injected at a concentration of 2–2.5 µg/µl. Electrodes (Nepagene CUY650P10) coated with highly conductive gel (Signagel, signa250) were positioned in the dorsoventral axis with the positive pole dorsal. Five electric pulses of 100 V, 50 ms pulse ON, 850 ms pulse OFF were applied using a Nepagene CUY21-SC electroporator. Pups were immediately warmed up in a heating chamber and brought to their cages at the end of the experiment.

## Demyelinating lesions

Before surgery, adult (2–3 months) WT mice were weighed, and an analgesic (buprenorphine, 30 mg/g) was administered to prevent postsurgical pain. The mice were anesthetized by induction of isoflurane (ISO-VET). Ocrygel (Tvm) was put on their eye to prevent dryness and lidocaine in cream (Anesderm 5%) was put on the ear bars to prevent pain. After cutting of the skin, a few drops of liquid lidocaine

were put to prevent pain. Focal demyelinating lesions were induced by stereotaxic injection of 1 µl of lysolecithin solution (LPC, Sigma, 1% in 0.9% NaCl) into the corpus callosum (CC; at coordinates: 1 mm lateral, 1.3 mm rostral to bregma, 1.7 mm deep) using a glass-capillary connected to a 10 µl Hamilton syringe. Animals were left to recover in a warm chamber before being returned into their housing cages.

## Tissue processing

Postnatal mice were transcardially perfused with 15 ml (P14) or 25 ml (>P21) of 2% PFA freshly prepared from 32% PFA solution (Electron Microscopy Sciences, 50-980-495). Perfused brains were dissected out, dehydrated in 10% sucrose, followed by 20% sucrose overnight, and embedded in OCT (BDH) before freezing and sectioning (16 µm thickness) in a sagittal plane with a cryostat microtome (Leica).

## Magnetic-assisted cell sorting (MACS)

Dissociation of cortex and corpus callosum from mice brain was done using neural tissue dissociation kit (P) (Miltenyi Biotec; ref 130-093-231). Briefly, cortices were dissected from P7, P12, P14, or P21 mice and dissociated using a MACS dissociator (Miltenyi Biotec; ref 130-096-427) followed by filtration through a 70 µm cell strainer (Smartstainer; Miltenyi Biotec; ref 130-098-462). Myelin residues were eliminated from P12, P14, and P21 mice cortices during an additional step using the debris removal kit (Miltenyi Biotec; ref 130-090-101). Cells were suspended in a 0.5% NGS solution, then incubated with anti-PDGFRα or anti-O4 coupled-beads (Miltenyi Biotec; ref 130-094-543 and 130-096-670). Unbound bead-coupled antibodies were washed away by centrifugation, leaving bound cells that were sorted using MultiMACS Cell24 Separator Plus (Miltenyi Biotec; ref 130-098-637). Sorted cells were either plated in culture plates for in vitro cell study or centrifuged at 1200 rpm and used for Western blot analysis or ChIP-seq.

## Immunofluorescence staining and microscopy

Postnatal mouse brain cryosections were dried for 20 min at room temperature (RT), before adding the blocking solution (10% normal goat serum [NGS, Eurobio, CAECHVOO-OU] and 0.1% Triton X-100 in PBS) for 1 hr at RT. Primary antibodies were diluted (dilutions indicated in the Key ressource table) in the same blocking solution and incubated on the slices overnight at 4°C. After washing with 0.05% Triton X-100 in PBS, sections were incubated with secondary antibodies conjugated to Alexa Fluor-488, Alexa Fluor-594, and Alexa Fluor-647 (Thermo, 1:1000). Finally, cell nuclei were labeled with DAPI (1/10,000, Sigma-Aldrich, D9542-10MG), and slices mounted in Fluoromount-G (SouthernBiotech, Inc 15586276).

In Situ Cell Death detection kit (Roche, 12156792910) was used to do TUNEL experiment on P21 mouse brains. Briefly, tissues were processed as mentioned above with anti-GFP and anti-CC1 and fixed in fixation solution for 20 min at RT. After washing, slices were permeabilized for 2 min in permeabilization solution (0.1% Triton X-100; 0.1% sodium citrate) and TUNEL reaction mixture was put on samples for 1 hr at 37°C. Tissues were then mounted in Fluoromount-G.

Fixed coverslips were blocked in blocking solution (10% NGS [Eurobio, CAECHVOO-OU] and 0.1% Triton X-100 in PBS) for 30 min at RT, incubated in the primary antibodies for 45 min at RT, and washed three times in 1× PBS. Secondary antibodies were applied for 45 min at RT and washed three times in 1× PBS. Coverslips were then incubated with DAPI solution for 5 min at RT. A final washing was done before mounting the coverslips on slides to be visualized under the microscope.

Immunofluorescence was visualized with Zeiss Axio Imager.M2 microscope with Zeiss Apotome system. Pictures were taken as stacks of 5–10 µm with 0.5 µm between sections. Image acquisition and processing were achieved by ZEN Microscopy and Imaging Software. Z-projections and orthogonal projections were done in ImageJ and processed with Adobe Photoshop. Figures were created using Adobe Illustrator.

## Western blot

Proteins from MACsorted cells were extracted during 30 min at 4°C in RIPA buffer (Thermo Fisher; 50 µl per million cells, 89901) supplemented with Halt Protease Inhibitor Cocktail (100×; Thermo Fisher, 87786). Protein concentration in the supernatant was estimated using the Pierce Detergent Compatible Bradford Assay Kit (Thermo Fisher, 23246). For each Western blot, we used 50 µg of

proteins denatured for 10 min at 95°C with added β-mercaptoethanol (from 24× stock) and Bolt LDS Sample Buffer (4×) (Thermo Fisher, B0007). Sodium dodecyl sulfate-polyacrylamide gel electrophoresis (SDS-PAGE) was performed using precast 4–12% polyacrylamide gradient gels (Thermo Fisher, NW04122BOX), submerged at 4°C in Bolt MOPS SDS Running Buffer (Thermo Fisher, B0001) using Mini Gel Tank and Blot Module Set (Thermo Fisher, NW2000). Precision Plus Protein All Blue protein standards (Bio-Rad, 1610373EDU) were run alongside the samples as a protein migration control. Proteins were separated for 90 min at 90 V, after which gels were transferred onto Amersham Protran 0.2 μm nitrocellulose membrane (Dutscher, 10600001) immersed at 4°C in NuPAGE Transfer Buffer (Thermo Fisher, NP0006-1) for 90 min at 60 V. Following transfer, membranes were incubated for 1 hr in TBS-T, 10% dry milk to aid blocking of nonspecific binding by the antibodies. Primary antibodies diluted in TBS-T were incubated with the membrane overnight at 4°C with shaking. After three washes in TBS-T, membranes were incubated with HRP-conjugated secondary antibodies diluted in TBS-T for 1 hr at 4°C with shaking, then developed using Pierce ECL Western Blotting Substrate (Thermo Fisher, 32109) and imaged with the ChemiDoc Touch Imaging System (Bio-Rad, 1708370). Western blot detection of actin was performed as loading control.

## Videomicroscopy

Tamoxifen was administered to P5 *Tns3^flox^; Pdgfrα-CreER^T^; Rosa26^stop-floxed-YFP^* and *Tns3^flox^; Pdgfrα-WT; Rosa26^stop-floxed-YFP^* littermates. Brains were dissected out at P7 in order to MACSort OPCs using an anti-PDGFRα antibody coupled to magnetic beads. OPCs were plated in poly-L-ornithine (Sigma, P4957)-coated μ-Slide 8 Well Glass Bottom slide (ibidi, 80827) at 40,000 cells/mm$^2$ in OPC proliferative medium: DMEM/F12 (Life Technologies, 31331028), 5 mM HEPES buffer (Life Technologies, 15630056), 0.6% glucose (Sigma, G8769), 1× penicillin/streptomycin (Life Technologies, 15140122), N2 supplement (Life Technologies, 17502048), B27 supplement (Life Technologies, 17504044), 20 ng/μl EGF (PeproTech, AF-100-15), 10 ng/μl FGF-basic (PeproTech, 100-18B), 10 ng/μl PDGF-AA (PeproTech, 100-13A), and 20 μg/ml insulin (Sigma, I6634). After 3 days of proliferation, medium was replaced by growth factor-depleted medium. Cell differentiation was tracked for 3 days using time-lapse video recording. Cells were put in to a videomicroscope (Zeiss AxioObserver 7, provided by ICM-quant and CELIS facilities) with a humidified incubator at 37°C with a constant 5% CO$_2$ supply. Images for both FITC and bright field were acquired every 10 min.

## Chromatin immunoprecipitation (ChIP)

ChIP-seq assays were performed as described previously (*Marie et al., 2018*) using iDeal ChIP-seq kit for Transcription Factors (Diagenode, C01010055). Briefly, O4$^+$ MACSorted cells were fixed in 1% formaldehyde (EMS, 15714) for 10 min at RT and the reaction was quenched with 125 mM glycine for 5 min at RT. Lysates were sonicated with a Bioruptor Pico sonicator (Diagenode, total time 8 min) and 4 μg of antibodies were added to sheared chromatin (from 4 million cells for Olig2 and from 1 million cells for histone marks) and incubated at 4°C overnight on 10 rpm rotation. Antibodies used were mouse anti-Olig2 antibody (Millipore, MABN50), rabbit anti-H3K4me3 antibody (Active Motif, 39060), rabbit anti-H3K27Ac antibody (Active Motif, 39034), rabbit anti-H3K4me1 antibody (Ozyme, 5326T), and mouse anti-H3K27me3 antibody (Abcam, ab6002). Mock (rabbit IgG) was used as negative control. Chromatin-protein complexes were immunoprecipitated with protein A/G magnetic beads and washed sequentially according to the manufacturer (Diagenode, C01010055). DNA fragments were then purified using IPure beads v2 (Diagenode, C01010055). Input (non-immunoprecipitated chromatin) was used as control in each individual experiment. The ChIP-seq libraries were prepared using Illumina TruSeq ChIP preparation kit and sequenced with Illumina NextSeq 500 platform.

## ChIP-seq analysis

All ChIP-seq analyses were done using the Galaxy Project (https://usegalaxy.org/). Reads were trimmed using Cutadapt (--max-n 4) and Trimmomatic (TRAILING 1; SLIDINGWINDOW 4 and cutoff 20; LEADING 20; MINLEN 50) and mapped using Bowtie2 onto mm10 mouse reference genome (-X 600; -k 2; `--sensitive`). PCR-derived duplicates were removed using PICARD MarkDuplicates. Bigwig files were generated with bamCoverage (binsize = 1). Peak calling was performed using MACS2 callpeak with Input as control and with options: `--qvalue 0.05`; `--nomodel`; `--keep-dup`

`1; --broad` (only for histone marks). Blacklisted regions were then removed using bedtools Intersect intervals.

Visualization of coverage and peaks was done using IGV (*Robinson et al., 2011*; http://software.broadinstitute.org/software/igv/home). Intersection and analysis of bound genes were done using Genomatix (https://www.genomatix.de/). Chd7, Chd8, and Mock ChIP-seq datasets are from *Marie et al., 2018*. Heatmap was done using R (4.0) and pheatmap package. GO analysis was done using Enrichr GO Biological Process 2021.

Two replicates were done for Olig2, with one of them of better quality (53,960 peaks for replicate 1 and 14,242 peaks for replicate 2). Only the peaks found in both replicates (6781) and the peaks from replicate 1 found in regulatory elements (13,948) were considered (16,578 in total). Three replicates were done for H3K4me3, two replicates were done for H3K27me3 and one replicate was done for H3K27Ac and H3K4me1. Intersection of these datasets was done using bedtools Intersect intervals.

Peaks overlapping with regions between 1000 bp upstream of transcription start site (TSS) and 10 bp downstream of TSS were identified as 'promoters' (Genomatix). 'Active promoters' were marked by peaks for H3K4me3 and H3K27Ac; 'Repressed promoters' by peaks for H3K27me3 and no active marks; 'Poised promoters' by peaks for H3K4me1 and no active or repressed marks. Regions outside promoters containing histone marks were considered as 'enhancers.' 'Active enhancers' were marked by peaks for H3K27Ac; 'Repressed enhancers' by peaks for H3K27me3 and no active marks; 'Poised enhancers' by peaks for H3K4me1 and no active or repressed marks. Genes were considered associated if the peaks were present in the promoter or within a range of 100 kb from the middle of the promoter and the gene expression was medium to high ('active'), low ('poised'), or not ('repressed') expressed (based on control RNA-seq dataset in GSE116601).

## RNA-seq analysis

Raw data were downloaded from GEO datasets GSE107919 and GSE116601 and processed through the Galaxy Project (https://usegalaxy.org/) using RNAstar for alignment on mm10 reference genome and featureCounts to obtain counts. Count per million (CPM), FPKM, and statistical analysis were done with R (4.0) using edgeR quasi-likelihood pipeline. Using control RNA-seq dataset in GSE116601, genes were classified based on their expression as not (below first quartile), low (between first quartile and mean), medium (between mean and third quartile), and high (above third quartile).

## scRNA-seq analysis

For mouse oligodendroglial cell analyses, counts per gene were downloaded from GEO datasets GSE75330 and GSE95194, and processed in R (4.0) using the following packages using de R-scripts deposited in https://github.com/ParrasLab/Tns3_paper_eLife_2022; *ParrasLab, 2022* and summarized here: *Seurat* (3.0) for data processing, *sctransform* for normalization, and *ggplot2* for graphical plots. Seurat objects were first generated for each dataset independently using *CreateSeuratObject* function (min.cells = 5, min.features = 100). Cell neighbors and clusters were found using *FindNeighbors* (dims = 1:30) and *FindClusters* (resolution = 0.4) functions. Clusters were manually annotated based on the top 50 markers obtained by the *FindAllMarkers*, adopting mainly the nomenclature from *Marques et al., 2016*. Using the *subset* function, we selected only the clusters containing neural progenitors and oligodendroglia cells. Using the *merge* function, we combined both oligodendroglial datasets into a single Seurat object (OLgliaDevPost). The new object was subjected to *NormalizeData*, *FindVariableFeatures*, *ScaleData*, *RunPCA*, and *RunUMAP* functions with default parameters. Different OPC clusters were fused into a single one keeping apart the cycling OPC cluster. For DimPlots and dot plots, clusters were ordered by stages of oligodendrogenesis from NSCs to myelinating OLs.

## Tns3$^{\beta geo}$ mice analysis

To explore the role of Tns3 in OL differentiation, we first analyzed a *Tns3* gene trap mouse line (*Tns3$^{\beta geo}$*) previously studied outside the CNS (*Chiang et al., 2005*), where the *βgeo* cassette is inserted after *Tns3* exon 4 (*Figure 3—figure supplement 1a*) driving *LacZ* transcription and by inserting a stop poly-A sequence, predicted to be a *Tns3* loss-of-function mutation. Despite the original report of postnatal growth retardation in *Tns3$^{\beta geo/\beta geo}$* mice, these mice were kept in homozygosity for several generations in C57BL/6 genetic background (Su-Hao Lo, UC Davis). We thus analyzed the impact in oligodendrogenesis in the postnatal brain of *Tns3$^{\beta geo}$* animals. We first immunodetected

βgalactosidase in OLs (Olig2⁺/CC1⁺ and Olig2⁺/PDGFRα− cells) of *Tns3^{βgeo}* postnatal brains at P21 (*Figure 3—figure supplement 1b and c*), paralleling our characterization of Tns3 expression with V5 and Tns3 antibodies. We then quantified the density of PDGFRα⁺ OPCs or CC1⁺ OLs in *Tns3^{βgeo/βgeo}* and *Tns3^{βgeo/+}* littermates at P21, finding similar number of OPCs and OLs in two main white matter areas (corpus callosum and fimbria; *Figure 3—figure supplement 1d, d', e*). Moreover, quantification of three different stages of OL differentiation (iOL1, iOL2, and mOL) by Olig2/CC1/Olig1 immuno-fluorescence did not reveal changes in the rate of OL differentiation (proportion of each stage) in *Tns3^{βgeo/βgeo}* mice compared to control littermates (*Figure 3—figure supplement 1f, f', g*). We veri-fied the homozygosity of *Tns3^{βgeo}* allele in *Tns3^{βgeo/βgeo}* mice by PCR amplification from genomic DNA of P21 brains finding that primers recognizing intron 4 and βgeo only produced PCR amplicons in *Tns3^{βgeo/βgeo}* mice but not when using intron 4 flanking primers that only produced PCR amplicons in wild-type mice (*Figure 3—figure supplement 1h*). We then checked for *Tns3* full-length transcripts using cDNA generated from P21 brains, and to our surprise, primers flanking exons 17 and 31 were similarly amplified from cDNA of *Tns3^{βgeo/βgeo}* and wild-type brains (*Figure 3—figure supplement 1i*), suggesting that in the brain of *Tns3^{βgeo/βgeo}* mice *Tns3* full-length transcripts coding for Tns3 protein are still produced. Altogether, these results suggested that, unlike in other tissues (*Chiang et al., 2005*), the *Tns3^{βgeo}* allele does not lead to *Tns3* loss of function in the brain, likely through the generation of alternative spliced *Tns3* variants, and is thus not suitable for assessing *Tns3* function in the CNS.

## Analyses of *TNS3* alleles in the human population based in gnomAD project

To assess whether TNS3 is potentially required during human development, we explore for the pres-ence of *TNS3* gene variants in the human population using the gnomAD database containing 125,748 exomes and 15,708 whole-genome sequences from unrelated individuals (*Karczewski et al., 2020*; *Lek et al., 2016*). Homozygous predicted loss-of-function (pLoF) alleles of *TNS3* were not found, and heterozygous pLoF were greatly below the expected frequency (0.1 observed/expected ratio, with 90% CI of 0.05–0.19; and LOEUF of 0.19; *Figure 3—figure supplement 2a*; https://gnomad. broadinstitute.org), meaning that heterozygous loss-of-function variants of *TNS3* cause ~80% devel-opmental mortality, a rate similarly high to key neurodevelopmental genes such as *SOX10* (LOEUF = 0.21; *Figure 3—figure supplement 2b*), *CHD7* (LOEUF = 0.08; *Figure 3—figure supplement 2c*), and *CHD8* (LOEUF = 0.08; *Figure 3—figure supplement 2d*), contrary to less broadly required factors such as *NKX2-2* (LOEUF = 0.67; *Figure 3—figure supplement 2e*) and *OLIG1* (LOEUF = 1.08; *Figure 3—figure supplement 2f*). Therefore, *TNS3* loss-of-function variants are badly tolerated in both mouse and human development.

## CRISPR/Cas9 tools development

CRISPOR software (http://crispor.tefor.net/) was used to design gRNAs with predicted cutting effi-ciency and minimal off-target and PCR amplification primers. The validation of *Tns3*-targeting CRISPR/Cas9 system was performed in 3T3 cells by transfection with Lipofectamine 3000 of *PX459* plasmids containing four different sgRNA sequences. After 2 days incubation, puromycin was added to medium for 4 days allowing survival of cells containing the *PX459* plasmid. Three days after proliferation in fresh medium without puromycin, DNA was extracted using DNeasy blood & tissue kit (QIAGEN). The target DNA for 5′ *Tns3* region was amplified by PCR using primers with the following sequences: forward: 5′-AGG TGG CCT TCA GCT CAGT-3′, reverse: 5′-GCT ATC ATC CCC ACT CAC CA-3′; annealing temperature of 64°C, with the PCR product expected to be 326 bp. DNA from 3′ *Tns3* target region was amplified using primer with the following sequences: forward: 5′-CCA GTC AGT GGT GAC ATT GTTT-3′, reverse: 5′-ACT GTT CCC AGG TTG CTA TCAT-3′, annealing temperature of 58°C, with the PCR product expected to be 419 bp. Cutting efficiency of sgRNA was verified by T7 endonuclease I, following the beta protocol of IDTE synthetic biology for amplification of genomic DNA and detecting mutations (*Figure 3—figure supplement 3c*), and using PAGE (*Figure 3—figure supplement 3d*). In order to generate plasmids that will insert CRISPR tools into the genome of the transfected cells and lead to permanent expression of the targeting tools, the *PX458* (GFP) or *PX459* (Puromycin) plasmids were subcloned into a *Tol2*-containing sequence backbone (obtained from *Tol2-mCherry*-expressing plasmid kindly provided by Jean Livet, Institut de la Vision, Paris). Finally,

to induce more robust expression of the Cas9 protein and reporter genes, we substitute the CMV promoter for the stronger CAG promoter.

## *Tns3⁴ᵈᵉˡ* and *Tns3¹⁴ᵈᵉˡ* mice generation by CRISPR and analyses

To generate new *Tns3* knockout mouse using CRISPR/Cas9 technology by introducing loss-of-function mutations (indels) at the beginning of *Tns3* full-length coding sequence. We generated CRISPR integrative plasmids (**Figure 3—figure supplement 3a**) driving Cas9 expression and gRNAs targeting *Tns3* exon 6 at the levels of the first coding ATG using as control plasmids without the *Tns3*-targeting sequence of the gRNA. Strong cutting efficiency of two gRNAs was validated by lipofection of neural progenitors (**Figure 3—figure supplement 3b–f**). We then used these optimized tools to induce CRISPR-mediated *Tns3* mutations in mouse zygotes, generating and characterizing two mouse lines having small deletions (4-deletion and 14-deletion) after the first coding ATG of *Tns3* (**Figure 3—figure supplement 1j**), expected to cause frame shifts leading to *Tns3* loss of function. Remarkably, homozygous animals were found in reduced numbers compared to Mendelian ratios with many of them dying during embryonic development (**Figure 3—figure supplement 1k**) with most homozygous animals showing major growth retardation by the second postnatal week compared with their littermates (**Figure 3—figure supplement 1l**), similar to the original report of *Tns3ᵝᵍᵉᵒ* mice (**Chiang et al., 2005**). Furthermore, we could still immunodetect Tns3 protein in CC1⁺ OLs of these homozygous mice at P21 with at least two different Tns3 antibodies (**Figure 3—figure supplement 1m and n**) and detect *Tns3* exons corresponding to *Tns3* full-length transcript by qPCR (**Figure 3—figure supplement 1o**). Further analysis of these mice was prevented by the Covid-19 lockdown, leading to the loss of these *Tns3* knockout mouse lines. Altogether, these results suggest that mice carrying constitutive *Tns3* loss-of-function mutations seems to escape the full *Tns3* loss of function in the brain by generating alternative spliced variants containing the main *Tns3* full-length exons, and thus we considered these animals not suitable to study *Tns3* function in oligodendrogenesis.

## RNA sequencing and analysis

Cortices from 3 to 4 animals for each group were dissected and frozen in liquid nitrogen for further processing. Total RNA was isolated with the TRIzol Reagent protocol (Thermo Fisher) and RNeasy Mini Kit (QIAGEN) according to the instructions of the provider. The RNA-seq libraries were prepared using either the NEBNext Ultra II Directional RNA Library Prep Kit (NEB) and sequenced with the NovaSeq 6000 platform (Illumina, 32 * 106 100-bp pair-end reads per sample). Quality of raw data was evaluated with FastQC. Poor quality sequences were trimmed or removed with fastp tool, with default parameters, to retain only good quality paired reads. Illumina DRAGEN bio-IT Plateform (v3.6.3) was used for mapping on mm10 reference genome and quantification with gencode vM25 annotation gtf file. Library orientation, library composition, and coverage along transcripts were checked with Picard tools. The following analysis was conducted with R software. Data were normalized with edgeR (v3.28.0) bioconductor packages prior to differential analysis with glm framework likelihood ratio test from edgeR package workflow. Multiple hypothesis-adjusted p-values were calculated with the Benjamini–Hochberg procedure to control FDR. For the differential expression analyses, low-expressed genes were filtered, sex was used as covariable (when relevant), and the cutoffs applied were FDR < 0.05. Finally, GO enrichment analysis of biological processes of the DEGs was conducted with clusterProfiler R package (v3.14.3).

## Statistical analysis

Experimental data is the result of optimization and analyses of several experiments done for each section of the study. Biological replicates (one sample comes from one animal) were used. Some quantifications concerning comparisons between *Tns3* loss-of-function genotypes and controls were done blindly.

Statistical parameters including the exact value of n, the definition of center, dispersion, and precision measures (mean ± SEM) and statistical significance are reported in the figures, figure legends, and **Source data 1**. 'n' represents the number of animals in histological studies and number of samples in RNA-seq, ChIP-seq, and ATAC-seq studies. Data distribution was assumed to be normal, but this was not formally tested. Statistical significance was determined using two-tailed Student's *t*-tests. One-way ANOVA test was performed for multiple comparisons or pairwise comparisons following

Turkey's ranking tests when comparing multiple groups. Data are judged to be statistically significant when p<0.05. In figures, asterisks denote statistical significance as calculated by Student's *t*-test (*p<0.05; **,p<0.01; ***p<0.001). No statistical methods were used to predetermine sample sizes, but our sample sizes are similar to those generally employed in the field to balance experimental robustness with the 3R rule for animal experimentation. Quantifications were performed from at least three independent experiments. No randomization was used to collect all the data, but they were quantified blindly. Statistical analysis was performed in Prism software.

## Data resources

Raw data files generated in this study have been deposited in the NCBI Gene Expression Omnibus under accession number GEO GSE203295 (https://www.ncbi.nlm.nih.gov/geo/query/acc.cgi?acc=GSE203293).

R-scripts used to treat RNA-seq data are deposited in GitHub (https://github.com/ParrasLab/Tns3_paper_eLife_2022, copy archived at swh:1:rev:fa63c93277571bd0fe113242e4929ea5a1957fad; *ParrasLab, 2022*).

## Contact for reagent and resource sharing

Further information and reasonable requests for reagents may be directed to and fulfilled by the corresponding author Carlos Parras (carlos.parras@icm-institute.org).

## Acknowledgements

We thank Dwight Bergles for the *PDGFRα::CreER*[T] mice and Jean Livet for Tol2-mCherry plasmid. Mathilde Bertrand for sharing ChIP-seq analysis pipeline. All animal work was conducted at the ICM PHENOPARC Core Facility. Data generated relied on ICM Core Facilities: PHENO ICMice, iGenSeq, DAC, iVector, CELIS, Histomics, and ICM Quant, and we thank all personnel involved for their contribution and help. The Core Facilities were supported by the 'Investissements d'avenir' (ANR-10-IAIHU-06 and ANR-11-INBS-0011-NeurATRIS) and the 'Fondation pour la Recherche Médicale.' This work was supported by funding by grants from the National Multiple Sclerosis Society (NMSS RG-1501-02851), and the Fondation pour l'Aide à la Recherche sur la Sclérose en Plaques (ARSEP 2014, 2015, 2018, 2019, 2020). EM, HH, and CM were supported by funding from Sorbonne Université. CM was also supported by Fondation pour la Recherche Médicale (FRM, FDT20160435662) and ARSEP grant 2018-2020.

## Additional information

### Funding

| Funder | Grant reference number | Author |
| --- | --- | --- |
| National Multiple Sclerosis Society | NMSS RG-1501-02851 | Carlos Parras |
| Fondation pour l'Aide à la Recherche sur la Sclérose en Plaques | ARSEP 2014 2015 2018 2019 2020 | Corentine Marie |
| Multiple Sclerosis Society | | Carlos Parras Corentine Marie Emeric Merour |
| Fondation pour la Recherche Médicale | | Corentine Marie |
| Sorbonne Universite | | Emeric Merour Hatem Hmidan Corentine Marie Adrien Clavairoly |

The funders had no role in study design, data collection and interpretation, or the decision to submit the work for publication.

## Author contributions
Emeric Merour, Formal analysis, Investigation, Visualization, Methodology, Writing - original draft, Writing - review and editing; Hatem Hmidan, Conceptualization, Formal analysis, Visualization, Methodology; Corentine Marie, Data curation, Visualization, Methodology, Writing - review and editing; Pierre-Henri Helou, Haiyang Lu, Antoine Potel, Jean-Baptiste Hure, Formal analysis, Investigation; Adrien Clavairoly, Conceptualization; Yi Ping Shih, Investigation; Salman Goudarzi, Formal analysis, Methodology; Sebastien Dussaud, Philippe Ravassard, Methodology; Sassan Hafizi, Conceptualization, Methodology; Su Hao Lo, Investigation, Methodology; Bassem A Hassan, Funding acquisition, Writing - review and editing; Carlos Parras, Conceptualization, Resources, Data curation, Software, Formal analysis, Supervision, Funding acquisition, Validation, Investigation, Visualization, Methodology, Writing - original draft, Project administration, Writing - review and editing

## Author ORCIDs
Sebastien Dussaud ⓘ http://orcid.org/0000-0002-5365-8338
Sassan Hafizi ⓘ http://orcid.org/0000-0002-4539-0888
Su Hao Lo ⓘ http://orcid.org/0000-0002-2675-9387
Bassem A Hassan ⓘ http://orcid.org/0000-0001-9533-4908
Carlos Parras ⓘ http://orcid.org/0000-0003-0248-1752

## Ethics
All animal procedures were performed according to the guidelines and regulations of the Inserm ethical committees (authorization #A75-13-19) and animal experimentation license A75-17-72.

## Decision letter and Author response
Decision letter https://doi.org/10.7554/eLife.80273.sa1
Author response https://doi.org/10.7554/eLife.80273.sa2

# Additional files

## Supplementary files
• MDAR checklist

• Source data 1. Statistics summary table.

• Supplementary file 1. Tables containing Olig2/Chd7/Chd8 ChIP-seq analyses and Tns3-iKO RNAseq analyses.

## Data availability
Sequencing data have been deposited in GEO under accession code GSE203295.

The following dataset was generated:

| Author(s) | Year | Dataset title | Dataset URL | Database and Identifier |
|---|---|---|---|---|
| Merour E, Marie C, Parras C | 2022 | Transient regulation of focal adhesion via Tensin3 is required for nascent oligodendrocyte differentiation | https://www.ncbi.nlm.nih.gov/geo/query/acc.cgi?acc=GSE203295 | NCBI Gene Expression Omnibus, GSE203295 |

The following previously published datasets were used:

| Author(s) | Year | Dataset title | Dataset URL | Database and Identifier |
|---|---|---|---|---|
| Marques S, Castello-Branco G | 2016 | scRNA-seq postnatal oligodendroglia | https://www.ncbi.nlm.nih.gov/geo/query/acc.cgi?acc=GSE75330 | NCBI Gene Expression Omnibus, GSE75330 |

*Continued on next page*

*Continued*

| Author(s) | Year | Dataset title | Dataset URL | Database and Identifier |
|---|---|---|---|---|
| Marques S, Castello-Branco G | 2018 | scRNA-seq developmental oligodendroglia | https://www.ncbi.nlm.nih.gov/geo/query/acc.cgi?acc=GSE95194 | NCBI Gene Expression Omnibus, GSE95194 |
| Miller KJ | 2021 | snRNA-seq human midgestation cerbellum | https://www.ncbi.nlm.nih.gov/projects/gap/cgi-bin/study.cgi?study_id=phs001908.v2.p1 | dbGaP, phs001908.v2.p1 |
| Zack DJ, Chamling X | 2021 | scRNA-seq from iPSC-derived Human Oligodendrocyte Progenitor Cells | https://www.ncbi.nlm.nih.gov/geo/query/acc.cgi?acc=GSE146373 | NCBI Gene Expression Omnibus, GSE146373 |

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
