## [Editor Report]

This work provides a major contribution to the field of oligodendrogenesis. It is a comprehensive analysis of Tensin3 function and is conceptually novel as it links the fields of transcriptional control in oligodendrocyte lineage cells with morphogenetic changes and integrin signaling-mediated cell survival. Finally, it is also the discovery of a very useful marker for remyelinating oligodendrocytes in disease conditions.

---

## [Decision Letter]

[Editors' note: this paper was reviewed by Review Commons.]

---

## [Author Response]

We thank all reviewers for their positive and encouraging comments and criticisms to improve our work. Here we present a reviewed version of the manuscript according to the comments risen.

Reviewer #1 (Evidence, reproducibility and clarity (Required)):This is an interesting paper that identifies Tns3 as a potential effector of oligodendrocytes differentiation based on an ingenious strategy comparing regulatory binding sites of known master regulators of differentiation, and then shows using in vivo genetics that this role is indeed correct. Next, a potential mechanism is identified by showing co-localization with β 1 integrin, known to regulate apoptosis of newly-formed oligodendrocytes. The results are well illustrated and the experiments performed with appropriate power using a broad range of techniques that combine in silico, in vitro and in vivo work to great effect.I think this represents an important contribution that will be of significant interest to neuroscientists – the mechanisms regulating oligodendrocytes generation remain poorly understood and the evidence that this contributes to adult learning (adaptive myelination) and CNS regeneration makes this a key question. I would suggest that the following are considered before publication:

We thank the reviewer for this positive comments and critics to improve the manuscript.

The work describing the KO mice that were not used as they proved unsuitable need not be described – it breaks the logical flow.

In agreement with the reviewer comment, we have reduced this part to a sort paragraph indicating that our analyses of several Tns3 constitutive KO lines showed developmental lethality and possible genetic compensation in Tns3 expression, leading us to conclude them inappropriate tools to study Tns3 function in oligodendrogenesis. We have summarized the data in Figure S7 and the description in the method section.

It would be useful to compare the extent of cell death in the Tns3 cKO mice with that described in the alpha6 integrin KO and the integrin beta1 cKO (the Colognato and Benninger papers). Do they match? If not (and I suspect the Tns3 cKO death is greater) could other mechanisms be downstream of the Tns3?

In agreement with the reviewer comment, we have added the following paragraph to the discussion:

‘Knockout mice for *integrin-*α*6* present a 50% reduction in brainstem MBP^+^ OLs at E18.5, just before they die at birth, accompanied by an increase in TUNEL^+^ dying OLs (Colognato et al., 2002). Similarly, conditional deletion of *integrin-*β*1* in immature OLs by *Cnp-Cre* also leads to a 50% reduction in cerebellar OLs at P5, with a parallel increase in TUNEL^+^ dying OLs (Benninger et al., 2006). Therefore, given that *Tns3*-induced deletion in postnatal OPCs also leads to 40-50% reduction in OLs in both grey and white matter regions of the postnatal telencephalon (this study), paralleled by similar increase in TUNEL^+^ apoptotic oligodendroglia, we suggest that Tns3 is required for integrin-β1 mediated survival signal in immature oligodendrocytes.’

I'm not sure why the authors argue that the activation of β 1 would not be informative experiment? This will regulate actin dynamics just as it regulates other integrin signaling pathways.Indeed, I would argue that an integrin activation experiments would be a neat way to prove mechanism (as it would be predicted to rescue the Tns3 cKO phenotype).

In agreement with the reviewer comment, we have removed this sentence: ‘If so, exogenous activation of integrin α6β1 in cultured OPCs by Mn^2+^ (Colognato et al., 2004) would not be expected to increase oligodendrogenesis in Tns3-iKO oligodendroglia.’

In an effort, to understand Tns3 function by acute Tns3-deletion in postnatal OPCs, we have compared the transcriptome of *Tns3-iKO* oligodendroglia compared to control cells, and we present these results in figure 7 pinpointing deregulated genes leading to reduced oligodendroglial differentiation, integrin dysregulation, increase apoptosis, and conflicting cell cycle signaling, and leaving for further studies the full characterization how the loss of Tns3 leads to the deregulation of these processes.

Can the authors provide any data on GM oligos and their OPCs? Is the requirement for Tns3 the same, and if so what might the implications be in the adult where new oligodendrocytes are being generated throughout life?

Indeed, in our analyses of *Tns3-iKO* mice, we provide quantifications of the cortex as a grey matter territory, showing a similar 40-50% reduction in OLs as in white matter areas corpus callosum and fimbria, and mixed regions such as the striatum.

I note in S13 that integrin beta1 is not highly expressed in human oligos at the time in question. Does this call into question the relevance for human disease?

We realize that scRNAseq plots are never easy to interpret but it is important to note that the levels of expression are coded by the intensity of the color scale, while the surface of the dot plots indicate the experimental sensitivity to detect transcript expression in a larger or smaller proportion of the cells in a given cluster/cell type (due to the drop out limitation of current single cell RNA-seq technologies). Considering this, please note that beyond a stronger expression in neural progenitor cells (NPCs, blue color), integrin-b1 (*Itgb1*) transcripts are expressed at medium to high levels (green to blue) in human immature OLs (Figure S13B), similar to their pattern of expression in mouse oligodendroglia (Figure S13A).

Reviewer #1 (Significance (Required)):See aboveReviewer #2 (Evidence, reproducibility and clarity (Required)):In this article, the authors identify and characterise Tensin3 (Tns3) as a target of key oligodendroglial transcription factors driving differentiation in the mouse. They use multiple transgenic models to describe loss of function, and suggest Tns3's action through integrin B1 signalling, with the key function being oligodendroglial survival.There is extensive and impressive work here, including identification of Tns3 by ChIPseq, expression of Tns3 in brain development, analysis of human (ES-derived) and mouse scRNAseq to infer timing of expression in the differentiation pathway, generation of V5-tagged Tns3-KI mice to overcome antibody limitations, identification of its expression in mouse remyelination, generation of a new Tns3KO mouse, in vivo Crispr Tns3KO in development, generation of a conditional KO, for deletion in adulthood, and finally some culture work to investigate potential mechanisms of actions.The bottom line is that Tns3 is required for survival of OPCs and immature oligodendrocytes in development/remyelination in mouse at least, and loss leads to apoptosis (through p53 increase and loss of integrin-B1 signalling), leading to a failure of proper differentiation.The experiments are carefully done, convincing and the tools generated impressive. There is clearly more to be done on clarifying the mechanism of action of Tns3, but I do not think further experiments on this topic are needed for this paper – they can wait for the next.

We thank the reviewer for the positive and encouraging reviewing comments. In an effort, to understand Tns3 function by acute Tns3-deletion in postnatal OPCs, we have compared the transcriptome of *Tns3-iKO* oligodendroglia compared to control cells, and we present these results in figure 7 pinpointing deregulated genes leading to reduced oligodendroglial differentiation, integrin dysregulation, increase apoptosis, and conflicting cell cycle signaling, and leaving for further studies the full characterization how the loss of Tns3 leads to the deregulation of these processes.

My only query is whether the expression of Tns3 is also in immature OLs in human brain (rather than human ES-derived OLs). This should be easily checked with interrogation of online Shiny apps from already published snRNAseq from various groups on human post mortem adult brain, but if not present then in also baby/fetal brain. This would be interesting and may well be different from the ES_derived cells which tend to be very immature and would add interest to the possible translational impact.

According to the suggestion of the reviewer, we analyzed 69,174 snRNAseq GW9-GW22 from fetal cerebellum,; Aldinger and Miller, 2021; https://doi-org.proxy.insermbiblio.inist.fr/10.1038/s41593-02100872-y, which we present now in Figure S3, finding a cluster of cells expressing iOL markers, including *NKX2-2, TNS3, ITPR2,* and *BCAS1,* similar to the hiPSCs-derived iOL1/iOL2 clusters and mouse iOL1/iOL2 clusters shown in Figure S2*.*

We also analyzed other datasets without finding iOLs given their age or numbers, including:

– Immunopanned PDGFRA+ cells from human cortex GW20-GW24 (2690 cells, Huang and Kriegstein, Cell 2020) finding OPCs but not iOLs.

– The recently published dataset from GW8-GW10 human forebrain oligodendroglia (van Brugen and Castelo-Branco, Dev Cell 2022; https://doi.org/10.1016/j.devcel.2022.04.016) containing OPCs but not iOLs.

– The GW17 to GW18 human cortex (40,000 cells, Polioudakis and Geschwind, 2019, https://doi.org/10.1016/j.neuron.2019.06.011) containing OPCs but not iOLs.

Reviewer #2 (Significance (Required)):This work extends our knowledge of oligodendroglial differentiation, links it to the ECM and provides interest in manipulating this in diseases including glioma.My expertise: myelin, oligodendroglia, remyelination, human neuropathologyReviewer #3 (Evidence, reproducibility and clarity (Required)):See belowReviewer #3 (Significance (Required)):Using purified oligodendrocytes target genes of key regulators of oligodendrocyte differentiation were analyzed, which led to the identification of Tensin-3. The authors performed a detail characterization of Tensin-3 expression. They found that Tensin-3 is highly expressed in immature mouse and human oligodendrocytes. Interestingly, Tensin-3 is selectively enriched in immature oligodendrocytes, and not present at detectable levels in OPCs and mature oligodendrocytes. Subsequently, the authors characterized Tensin-3 function by a series of knockdown approaches in vitro and in vivo. These series of experiments revealed an essential function of Tensin-3 in supporting oligodendrocytes survival. In the absence of Tensin-3 a large fraction of oligodendrocytes undergo apoptosis while differentiating to mature oligodendrocytes.This is a remarkable study applying an impressive array of methods that led to an important discovery in the field of oligodendrocyte biology. The main advances for the field are: (1) identification of a novel marker for premyelinating oligodendrocytes, (2) elucidation of Tensin-3 as a pro-survival factor in oligodendrocytes differentiation, (3) evidence of link of Tensin-3-integrin signal in survival of oligodendrocytes.The data is well presented and organized, and the paper well written. I recommend publication with only minor suggestions for a revision:

We thank the reviewer for this positive comments and critics to improve the manuscript.

In Figure 2, only images are shown, and the data is referred to as highly expressed or strong colocalization. Even if the data looks clear, the authors should provide some quantification of the data in the figure.

We thank the reviewer for his comment and we have now provided a quantification of the fraction of Tns3+ cells expressing different markers of oligodendrocyte lineage progression/stages, and the percentage of each stage expressing Tns3.

Figure 3 is given too much weight in the manuscript text. I would recommend to shorten the text in the result section, and to move this figure to the supplement as it does not advance the story. It mainly shows that the KO mice still express transcripts in the brain. Were the transcripts lost in peripheral tissue?

As mentioned above, in agreement with the reviewers #1 and #3 comments, we have reduced this part to a sort paragraph indicating that our analyses of several Tns3 constitutive KO lines showed developmental lethality and possible genetic compensation in Tns3 expression, leading us to conclude them inappropriate tools to study Tns3 function in oligodendrogenesis. We have summarized the data in Figure S7 and the description in the method section.

Page 11: the authors describe in the text how the floxed allele was generated. This should be shifted to the supplement.

According to reviewers suggestion, we have moved the description of Tns3 floxed allele generation to the Methods section.

Page 16: the authors refer to Bcas1 as a problematic marker for immature oligodendrocytes, because the transcript is also expressed in mature oligodendrocytes. The authors are correct that the transcript is expressed in mature oligodendrocytes. However, the proteins changes its localization when oligodendrocytes mature. On protein level, it is valuable and a selective marker, as antibodies only label pre-myelinating and actively myelinating cells. In mature oligodendrocytes, antibodies against Bcas1 do not label the cell, only myelin. The text is misleading and needs to be corrected.

In agreement with reviewers comment we have modified the text as follows: ‘An optimized protocol for immunodetection using Bcas1-recognizing antibodies has been shown to label iOLs (Fard et al., 2017).’